# VideoKR: Towards Knowledge- and Reasoning-Intensive Video Understanding

**Lin Fu** [1] [*]  **Zheyuan Yang** [2] [*]  **Yang Wang** [1]  **Tingyu Song** [3]  **Arman Cohan** [4]  **Yilun Zhao** [4]

🤗 VideoKR    ⦿ VideoKR

## Abstract

We introduce VideoKR, the first large-scale training corpus specifically designed to strengthen knowledge- and reasoning-intensive video understanding. It comprises 315K video reasoning examples over 145K newly collected, CC-licensed, expert-domain videos. We develop a human-in-the-loop, skill-oriented example generation pipeline that targets progressively deeper video reasoning capabilities while ensuring the difficulty, diversity, and reliability of both the examples and their CoT rationales. We also curate VideoKR-Eval, a new expert-annotated benchmark where questions require genuine video understanding and knowledge-intensive reasoning rather than textual shortcuts. Our experiments show that, under a standard SFT→GRPO pipeline, models post-trained on VideoKR outperform prior post-training approaches on knowledge-intensive video reasoning while remaining competitive on general video reasoning, highlighting data design as a key driver of progress in video reasoning. We further conduct comprehensive ablations to isolate the contributions of VideoKR, providing actionable insights for future work.

## 1. Introduction

Multimodal foundation models for video understanding have achieved rapid progress in recent years, driven by architectural advances (Shu et al., 2025; Zohar et al., 2025; Ren et al., 2025; Li et al., 2024d; 2026), large-scale pre-training (Zhang et al., 2025a; Wang et al., 2025c; Bai et al.,

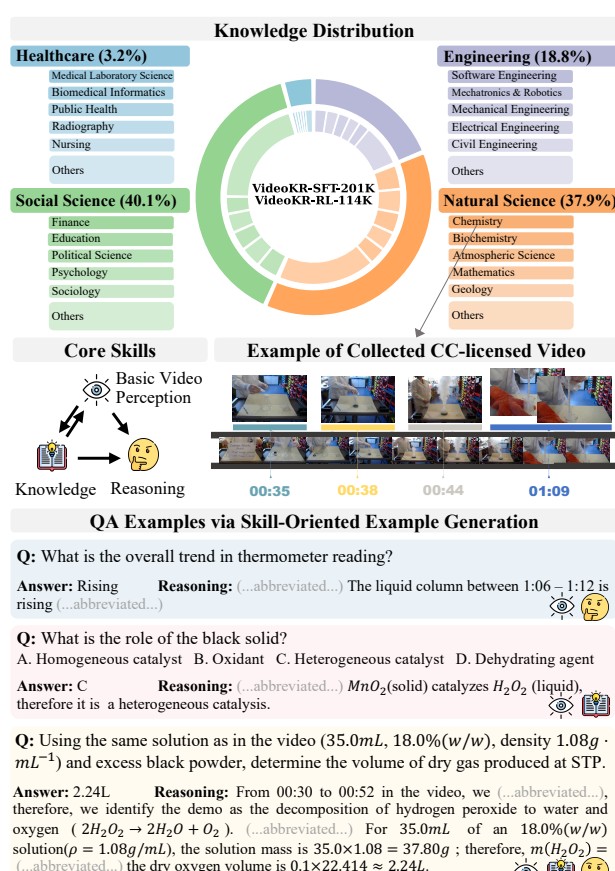

*Figure 1.* An overview of the VideoKR training corpus. All videos are newly collected and CC licensed, and span a wide range of professional domains. We develop a skill oriented QA synthesis pipeline in which every example is grounded in one of three core skills essential for advanced video reasoning, and examples in the CoT subset are further paired with a high quality reasoning trace.

2025; Chen et al., 2025a), and sophisticated post-training strategies (Feng et al., 2025a; Meng et al., 2025; Wang et al., 2025a; Li et al., 2025a;b). However, current models still face significant limitations when transitioning from surface-level video perception to video reasoning tasks that demand domain knowledge and multi-step inference (Zhao et al., 2025b; Hu et al., 2025; Deng et al., 2025; Song et al., 2025). A key bottleneck lies in the nature of the training corpora

---

[*]Equal contribution [1]Zhejiang University, Hangzhou, Zhejiang, China [2]Tongji University, Shanghai, China [3]University of Chinese Academy of Sciences, Beijing, China [4]Yale University, New Haven, CT, USA. Correspondence to: Yilun Zhao <yilun.zhao@yale.edu>.

*Proceedings of the 43rd International Conference on Machine Learning*, Seoul, South Korea. PMLR 306, 2026. Copyright 2026 by the author(s).

*Table 1.* Comparison of VideoKR with prior post-training corpora for video understanding. **%Video** denotes the fraction of video understanding examples, **CC** indicates whether all videos are Creative Commons (CC) licensed. [†]: data has not been open-sourced.

| | Video Source | %Video | # Video | Avg Duration | CC | # Example | Example Source | Expert-domain | Example/CoT Generator |
|---|---|---|---|---|---|---|---|---|---|
| LLaVA-Video (Zhang et al., 2025c) | Existing Dataset | 100% | 178K | 36.9 seconds | ✗ | 1156K | Newly Generated | ✗ | GPT-4o |
| VideoEspresso (Han et al., 2025) | Existing Dataset | 100% | 259K | 47.7 seconds | ✗ | 202K | Newly Generated | ✗ | GPT-4o |
| Video-R1 (Feng et al., 2025a) | Existing Dataset | 52% | 61K | 36.9 seconds | ✗ | 260K | Existing Dataset | ✗ | Qwen2.5-VL-72B-Instruct |
| VideoRFT (Wang et al., 2025a) | Existing Dataset | 56% | 127K | 24.7 seconds | ✗ | 310K | Existing Dataset | ✗ | GPT-4o-mini + DeepSeek-R1 |
| [†]Video-CoT (Zhang et al., 2025b) | Existing Dataset | 100% | Unknown | Unknown | ✗ | 192K | Existing Dataset | ✗ | Qwen2.5-VL-72B-Instruct |
| OneThinker (Feng et al., 2025b) | Existing Dataset | 42% | 158K | 90.9 seconds | ✗ | 600K | Existing Dataset | ✗ | Seed1.5-VL |
| VideoAuto-R1 (Liu et al., 2026a) | Existing Dataset | 59% | 35K | 63.8 seconds | ✗ | 83K | Existing Dataset | ✗ | – |
| **VideoKR (Ours)** | Newly Collected | 100% | 145K | 344.1 seconds | ✓ | 315K | Newly Generated | ✓ | Expert-validated selection from a pool of 7 frontier models |

used to develop these models.

Existing large-scale video datasets are predominantly constructed for perceptual objectives such as action recognition, event localization, and short-range temporal understanding (Feng et al., 2025a; Wang et al., 2025a; Meng et al., 2025; Chen et al., 2025b; 2026). Their content is heavily skewed toward everyday activities, with limited coverage of specialized domains and little support for knowledge- and reasoning-intensive video understanding. Consequently, models trained on current corpora often struggle with tasks requiring multi-hop inference, scientifically grounded explanations, or interpretation of events governed by non-observable principles, limiting their reliability in real-world applications that demand accurate, domain-aware reasoning.

To bridge this gap, we open-source VideoKR, the first large-scale training corpus targeted for knowledge- and reasoning-intensive video understanding. We collect 145K CC-licensed videos across 82 professional subjects using a *knowledge-driven* collection protocol that targets real-world manifestations of domain knowledge. To transform these raw videos into effective video reasoning training data, we design a *skill-oriented QA generation* framework that decomposes knowledge- and reasoning-intensive video understanding into three complementary capabilities: *basic video reasoning*, *knowledge-enhanced video perception*, and *knowledge-intensive video reasoning*. For each video, the framework generates challenging QA examples tailored to each skill category, each paired with a high-quality CoT rationale. We apply rigorous quality control with human-expert involvement, yielding a high-quality supervised fine-tuning corpus, VideoKR-SFT-201K, and a reinforcement learning corpus, VideoKR-RL-114K. In addition, through a manual audit of existing knowledge-intensive video reasoning benchmarks, we find that many examples are solvable with little video understanding. To address these issues, we construct a new evaluation benchmark, VideoKR-Eval.

We adopt a standard SFT→GRPO pipeline to isolate data design as the primary bottleneck and attribute performance gains more cleanly to VideoKR. We further establish a standardized evaluation framework to enable fair, reproducible

model comparisons. Experiments show that even without sophisticated post-training algorithmic design, base models (*i.e.,* Qwen2.5-VL-7B-Instruct and Qwen3-VL-8B-Instruct) post-trained on VideoKR already outperform prior post-training approaches. To inform how VideoKR can advance future work in video reasoning, we conduct comprehensive ablations that disentangle its key contributors, including the effect of CoT supervision, the impact of skill-based data composition, and controlled SFT and RL studies that compare VideoKR against prior post-training corpora.

We summarize our main contributions as follows:

- We open-source VideoKR, the first large-scale training corpus designed for knowledge- and reasoning-intensive video understanding. We apply rigorous quality control to ensure consistently high-quality training data (§3).

- We construct VideoKR-Eval, a new evaluation benchmark that mitigates single-frame answerability in prior benchmarks through multi-model single-frame probing and expert re-annotation of filtered videos (§4).

- We establish a standardized evaluation framework to ensure fair and reproducible model comparisons (§5.2).

- Models post-trained on VideoKR achieve the best knowledge-intensive performance among similar-sized models, while remaining competitive on general video benchmarks (§6).

- We conduct comprehensive ablations to isolate the contributions of VideoKR, including the effectiveness of CoT supervision, the impact of skill-based data composition, comparisons against prior post-training corpora, yielding actionable insights for future work (§6.4 to §6.5).

## 2. Related Work

**Video Understanding Datasets.** Recent video understanding benchmarks have expanded their scope to evaluate a broader range of multimodal and reasoning capabilities (Wu et al., 2021; Zhou et al., 2025; Fu et al., 2025; Zhao et al., 2025b; Deng et al., 2025; Li et al., 2024c; Liu et al., 2026b; Xu et al., 2025). General-purpose benchmarks such as Video-MME (Fu et al., 2025), MVBench (Li

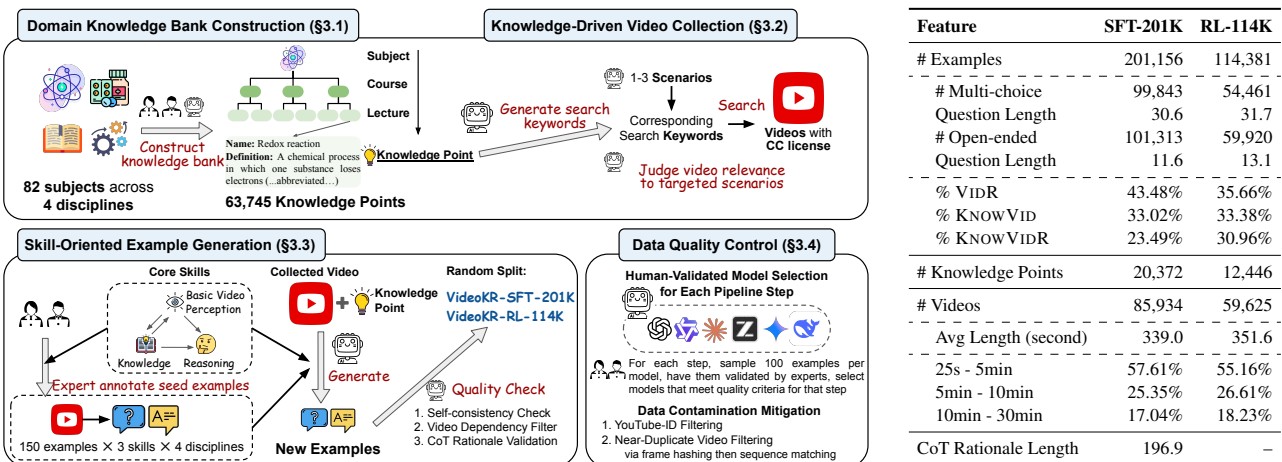

| Feature | SFT-201K | RL-114K |
|---|---|---|
| # Examples | 201,156 | 114,381 |
| # Multi-choice | 99,843 | 54,461 |
| Question Length | 30.6 | 31.7 |
| # Open-ended | 101,313 | 59,920 |
| Question Length | 11.6 | 13.1 |
| % VIDR | 43.48% | 35.66% |
| % KNOWVID | 33.02% | 33.38% |
| % KNOWVIDR | 23.49% | 30.96% |
| # Knowledge Points | 20,372 | 12,446 |
| # Videos | 85,934 | 59,625 |
| Avg Length (second) | 339.0 | 351.6 |
| 25s - 5min | 57.61% | 55.16% |
| 5min - 10min | 25.35% | 26.61% |
| 10min - 30min | 17.04% | 18.23% |
| CoT Rationale Length | 196.9 | – |

*Figure 2.* (Left) Overview of data construction pipeline. (Right) Statistics of VideoKR-SFT-201K and VideoKR-RL-114K training corpus.

et al., 2024b), VSI-Bench (Yang et al., 2025b), and Video-Vista (Li et al., 2024c) assess perceptual skills, spatiotemporal comprehension, and cross-modal reasoning, providing a solid foundation for evaluating video understanding. Building on this trend, a growing set of knowledge-, science-, and reasoning-intensive evaluation benchmarks focuses on deeper, domain-aware reasoning that goes beyond surface-level video perception (Xu et al., 2025; Liu et al., 2026b). For instance, MMVU (Zhao et al., 2025b) requires models to reason over specialized-domain videos and apply relevant domain knowledge; VideoMMMU (Hu et al., 2025) and Video-MMLU (Song et al., 2025) target expert-level understanding of subject-specific lecture videos; and SciVideoBench (Deng et al., 2025) evaluates advanced reasoning over scientific videos.

**Post-training for Video Understanding.** Current reasoning models are typically trained through a two-stage post-training pipeline that combines SFT and RL (Guo et al., 2025; Team et al., 2025; Yang et al., 2025a; Li et al., 2025c). To enhance video reasoning capabilities, the SFT stage is usually initialized on video reasoning datasets that include explicit chain-of-thought annotations, temporal cues, and spatial grounding signals, helping the model form more structured and interpretable reasoning patterns (Munasinghe et al., 2025; Zohar et al., 2024; Wu et al., 2025; Feng et al., 2025b; Wang et al., 2025b). In the RL phase, recent work has concentrated on adapting reinforcement learning with verifiable rewards (RLVR) to video reasoning, exploring complex reward engineering that emphasizes spatial understanding (Ouyang et al., 2025; Tang et al., 2025a), temporal dynamics (Wang et al., 2025d; Chen et al., 2025b; Feng et al., 2025a; Zhao et al., 2025a), or integrated spatiotemporal relationships (Meng et al., 2025; Tang et al., 2025b). Despite these advances, most post-training approaches still build on repurposed video understanding datasets that target

basic perception: as shown in Table 1, existing open-source corpora largely rely on short videos from datasets released years ago, and synthesis-based efforts (Zhang et al., 2025c; Han et al., 2025) typically depend on a single model, which can introduce systematic biases. We address this need by open-sourcing VideoKR: every video is newly collected, depicts expert-domain scenarios, and is released under CC licenses, and we adopt a human-in-the-loop, skill-oriented example generation framework to ensure the difficulty, diversity, and reliability of the data. Our experiments show that, under a standard SFT→GRPO pipeline, models post-trained on VideoKR already outperform prior post-training approaches, suggesting that better data remains crucial.

## 3. VideoKR Training Corpus Construction

This section describes the VideoKR data construction process, with an overview of the pipeline shown in Figure 2. Because our goal is large-scale corpus synthesis, exhaustive manual construction is infeasible. However, model based generation can introduce systematic artifacts. We therefore adopt a quality-controlled, semi-automated pipeline: whenever a step involves model outputs, it is audited and validated by human experts (detailed in Section 3.4). We engage 34 domain experts, each with graduate-level background in the relevant discipline (see Appendix A.2 for annotator biographies), to enforce quality criteria throughout the process.

### 3.1. Domain Knowledge Bank Construction

To achieve comprehensive coverage of domain-related videos, we begin by constructing a *Domain Knowledge Bank*, where each entry represents a *knowledge point* consisting of a term and its corresponding definition. The authors manually reviewed undergraduate curricula from top universities worldwide and identified 82 representative subjects distributed across four major disciplines: Natural Sci-

ences, Healthcare, Humanities and Social Sciences, and Engineering. The complete subject list is provided in Appendix A.1. To ensure systematic and fine-grained representation of domain knowledge within each discipline, we adopt a hierarchical knowledge organization framework with four layers: *Subject → Course → Lecture → Knowledge Point*. Specifically, for each subject, we ask expert annotators to provide a list of 4 to 8 core undergraduate courses consistent with standard academic programs. For each selected course, annotators compile a structured syllabus based on well-established curricula from top universities, outlining major lecture topics and learning objectives. Then for each lecture, we prompt LLMs (see Section 3.4 for our expert-involved procedure to select and validate the models used here and in subsequent steps) to identify several key knowledge points, each paired with a term and a paragraph-length definition. We collect a total of 63,745 knowledge points.

### 3.2. Knowledge-Driven Video Collection

Building upon the constructed domain knowledge bank, we then collect a large-scale video corpus, outlined below.

**Knowledge-based Video Scenario Generation.** In practice, directly using knowledge point terms (*e.g.,* "Newton's Second Law") as search queries often yields lecture recordings or purely instructional materials. While informative, such videos often lack the diversity and situational richness found in real-world contexts where the knowledge is implicitly applied. To obtain more authentic and engaging content, we first ask LLMs to generate 1–3 short scenarios that describe realistic situations involving each knowledge point. For example, instead of searching "Newton's Second Law", a generated scenario might be "a rocket launching into the sky", which inherently reflects Newton's Second Law. These scenarios are then transformed into semantically relevant search keywords, helping retrieve videos that embody the knowledge rather than merely explain it.

**Scenario-Guided Video Search and Filtering.** For each generated search keyword, we employ the YouTube Data API[1] to retrieve metadata (*e.g.,* titles, descriptions, and durations) for the top-10 candidate videos. We restrict the search to videos released under CC licenses to ensure legal reusability, a critical aspect that has been ambiguous in prior training corpora. For each candidate video, we instruct models to evaluate its relevance to the expected knowledge point and scenario based on the textual metadata. Videos exceeding 30 minutes are excluded, as long-context video understanding falls beyond the scope of this work. For the remaining candidates, we download the videos and prompt MLLMs to perform a secondary relevance assessment using visual content to confirm alignment with the intended knowledge context. To remove potentially harmful or sensitive

content, we randomly sample four frames from each video and run Azure AI's image moderation APIs[2] to filter out unsafe videos. We collect 146,567 CC-licensed videos.

### 3.3. Skill-Oriented Example Generation

For each video, we generate multiple QA examples. Following recent video post-training work (Feng et al., 2025a; Meng et al., 2025), we adopt *multi-choice* and *open-ended* QA formats, as they offer verifiable supervision suitable for RLVR. To enable scalable and high-quality data creation, we design a skill-based example generation pipeline:

**Core Skill Categorization.** As illustrated in Figure 1, three complementary dimensions, *i.e., perception*, *knowledge*, and *reasoning*, are essential for knowledge- and reasoning-intensive video understanding. Accordingly, we define three core skills that guide the example generation process: (1) *Basic Video Reasoning* (VIDR), which involves direct comprehension of events observable from the visual sequence, such as tracking actions, spatial relations, or temporal order, without relying on external domain knowledge. (2) *Knowledge-enhanced Video Perception* (KNOWVID), where visual perception is enriched by explicit domain knowledge. The model must align observed visual cues with relevant concepts across both spatial and temporal dimensions, for example, recognizing laboratory apparatus such as a "burette" or "condenser" and understanding their roles in a sequence of chemical procedures. (3) *Knowledge-Intensive Video Reasoning* (KNOWVIDR), which focuses on integrating visual understanding with domain knowledge to perform sophisticated, multi-hop inference, *e.g.,* estimating the quantity of chemical product formed from observed reactant amounts, or inferring a patient's likely diagnosis by interpreting symptoms and medical procedures depicted in a clinical video.

**Seed Examples Curation By Human Experts.** To ensure the quality and domain accuracy of the generated examples, we engage expert annotators to curate a seed set of examples for every core skill defined above. For each core skill, the annotators select representative knowledge points and their corresponding collected videos, then construct *question–answer* pairs accompanied by detailed, step-by-step *reasoning processes* that clearly articulate how visual evidence, domain knowledge, and logical inference jointly lead to the final answer. Each annotated example then undergoes a manual review by the authors to verify the accuracy of the QA content and reasoning process. In total, 150 examples are created for each skill within every discipline, resulting in 1,800 high-quality, expert-curated seed examples. To improve reliability, each example is independently reviewed by a second annotator; 74 examples are revised in this stage.

---

[1] https://developers.google.com/youtube/v3/docs.

[2] https://learn.microsoft.com/en-us/azure/ai-services/content-safety/quickstart-image.

**Example Generation.** Building on the expert-curated seed set, we use frontier MLLMs to scale up example creation in a controlled, skill-aware manner. For each video, the model generates two examples per skill, producing six examples in total through six independent generation rounds (one per example). During each round, the model is provided with (1) video frames uniformly sampled at 0.2 fps with timestamps, (2) three randomly sampled human-curated examples from the same discipline and skill category, and (3) the knowledge point and associated subjects when targeting the KNOWVID or KNOWVIDR skills. The model is instructed to emulate the seed examples in QA formulation and reasoning process generation, while maintaining fidelity to the video's unique visual content and knowledge context.

**Example Validation and Filtering.** To mitigate generation errors and reasoning bias, we adopt three complementary strategies: (1) *Self-Consistency Verification*: The model is re-prompted with the generated question and corresponding video frames to produce a detailed, step-by-step answer. An example is retained only if the re-derived answer matches the original, and the reasoning process from this verification step is used as the final reasoning trace. (2) *Video Dependency Filtering*: To ensure that each example in VideoKR genuinely requires visual understanding rather than relying on textual cues or shortcut reasoning, we instruct InternVL3.5-38B and Qwen3-VL-32B-Instruct to answer the question using only the text and four randomly sampled video frames. If both models successfully predict the correct answer under this limited setting, the example is removed from the dataset. Notably, this filter is stricter than what existing *evaluation* benchmarks use, which commonly rely on text-only (Fu et al., 2025; Di & Xie, 2024) or single-frame (Saravanan et al., 2025; Plizzari et al., 2025) settings. (3) *CoT Rationale Validation:* To mitigate systematic generator bias in the reasoning traces, we use an independent strong MLLM as a verifier: given the question, reasoning trace, and videos, it checks that each key step is supported by observable evidence or standard domain knowledge, and that the reasoning decisively distinguishes the chosen answer from plausible alternatives. We discard examples with critical unsupported steps.

### 3.4. VideoKR Data Quality Control

Beyond the automated example validation and filtering described above, we further strengthen VideoKR through pipeline-level quality control and contamination mitigation.

**Human-Validated Model Selection for Each Pipeline Step.** As summarized in Table 1, prior work on video reasoning corpus construction typically relies on a single model throughout the pipeline. This design can introduce model specific artifacts. To improve synthesis diversity and avoid overcommitting to any single model's biases, we use a pool of seven frontier models (*i.e.,* GPT-5.2, GPT-5-mini, Claude-4.5-Sonnet, Gemini-3-Flash, DeepSeek-V3.2, Qwen3-VL-235B-A22B, and GLM-4.6V). However, our pipeline is difficulty stratified: lightweight steps such as metadata relevance screening can be handled reliably by multiple models, whereas demanding stages such as QA example generation and verification are reliable only for a subset of models. We therefore introduce a human validated model selection protocol to determine stage eligibility: For each candidate model and each pipeline step, we sample 100 instances from that step's real inputs and ask domain expert annotators to assess the model outputs and label errors. A model is eligible for a step only if its error rate falls below a predefined threshold. We provide details of this process and the human-validated models for each step in Appendix A.4. During large-scale synthesis, for each instance at a given step, we randomly select one qualified model from the pool.

**Data Contamination Mitigation.** To prevent evaluation leakage, we traverse all video benchmarks supported by LMMs-Eval (Li et al., 2024a) and apply a two-stage decontamination protocol over the videos in VideoKR: (1) *YouTube-ID Filtering*: When a benchmark (*e.g.,* MMVU) provides YouTube video IDs, we directly filter out any training video whose YouTube ID matches an evaluation video ID, resulting in 131 videos removed. (2) *Near-Duplicate Video Filtering*: We also perform duplicate detection using frame level perceptual hashing and windowed sequence matching, resulting in 877 videos removed. We detail the process in Appendix A.5.

**Manual Quality Assessment.** To further assess the end-to-end quality of the finalized corpus and quantify residual noise, we randomly sample 800 examples from VideoKR-SFT-201K and ask ten expert annotators, who previously participated in seed-example curation, to evaluate them end to end. Of the sampled items, 52 questions are flagged as potentially non visual solvable. For reasoning traces, annotators identify 32 errors. Among these, 17 cases change the final answer, while 15 cases preserve the correct answer but rely on unsupported domain claims or fail to ground key steps in the relevant video evidence. These error rates are comparable to the error levels observed during human expert seed example curation (§3.3) and are therefore acceptable.

### 3.5. VideoKR-SFT-201K & VideoKR-RL-114K

We then randomly partition the generated 315,537 examples into two subsets while preserving video-level grouping, resulting in VideoKR-SFT-201K for supervised fine-tuning and VideoKR-RL-114K for RLVR training. For VideoKR-SFT-201K, each example retains its validated CoT rationale as the supervision target, whereas VideoKR-RL-114K keeps only the question and verifiable answer, since RLVR optimizes against the verifiable answer while the policy model generates its own reasoning during training. Figure 2 presents the key data statistics. Randomly-sampled

*Table 2.* Single-frame answerability rates across existing benchmarks. A QA example is classified as single-frame-solvable for a model only if the model answers it correctly in all three independent trials using only the question, answer options, and one randomly sampled video frame.

| Model | VidMMMU (900) | MMVU (1,000) | SciVidBench (1,000) | VideoKR-Eval (ours) (2,000) |
|---|---|---|---|---|
| Claude-4.5-Sonnet | 35.3 | 41.3 | 21.8 | 9.5 |
| Qwen3-VL-235B-A22B | 39.3 | 45.2 | 13.2 | 10.1 |
| GPT-5.2 | 38.3 | 49.7 | 23.0 | 10.7 |

examples from VideoKR are shown in Appendix A.3.

## 4. VideoKR-Eval Evaluation Benchmark

We next discuss the motivation and build of VideoKR-Eval.

### 4.1. Limitations of Existing Benchmarks

We observe that existing knowledge-intensive video reasoning benchmarks (*e.g.,* VideoMMMU, MMVU, and SciVideoBench) contain a substantial fraction of examples that can be answered without continuous video understanding. We quantify this issue through single-frame probing: each model is given only the question, answer options, and one randomly sampled frame from the video, and each example is evaluated in three independent trials. As shown in Table 2, frontier models achieve surprisingly high single-frame answerability rates on existing benchmarks (*e.g.,* >35% on MMVU and VideoMMMU).

### 4.2. VideoKR-Eval Benchmark Construction

We construct **VideoKR-Eval** from VideoMMMU, MMVU, and SciVideoBench by retaining 1,254 original examples that require continuous video understanding under multi-model single-frame probing, and augmenting them with 746 expert-reannotated examples from filtered videos.

**Multi-Model Single-Frame Filtering.** For each example in VideoMMMU, MMVU, and SciVideoBench, we run single-frame probing with three frontier models: Qwen3-VL-235B-A22B, Claude-4.5-Sonnet, and GPT-5.2. Each model receives only the question, answer options, and one randomly sampled frame from the video, and is evaluated with three independent trials. For each model, an example is considered single-frame-solvable if the model answers it correctly in all three trials; otherwise, it is treated as requiring continuous video understanding for that model. We retain only the intersection of examples judged as requiring continuous video understanding by all three models, yielding 1,254 original examples.

**Expert Re-annotation of Filtered Videos.** For examples outside this intersection, we discard the original QA pairs and ask domain experts to re-annotate new QA examples

using the corresponding videos. Annotators are required to write questions grounded in clearly observable video evidence, requiring relevant domain knowledge, and paired with uniquely determined ground truth answers. This process yields 746 expert-reannotated examples. Together with the 1,254 retained original examples, VideoKR-Eval contains 2,000 examples. Detailed statistics for VideoKR-Eval are provided in Appendix B.1.

## 5. Experiment Setup

In this section, we discuss the experiment setup for post-training on VideoKR and the subsequent model evaluation.

### 5.1. Post-Training on VideoKR

Recent post-training work for video reasoning emphasizes sophisticated RL variants and reward engineering. In contrast, we aim to isolate a different, and arguably more fundamental, bottleneck: *whether data design is the primary limiting factor for knowledge and reasoning intensive video understanding.* Accordingly, we deliberately adopt a standard, widely used SFT→GRPO pipeline as a controlled scaffold, ensuring that algorithmic complexity does not become a confounder and that observed gains can be attributed more cleanly to the training data.

Specifically, we use Qwen2.5-VL-7B-Instruct and Qwen3-VL-8B-Instruct as base models to assess whether VideoKR yields consistent gains under various architectural designs and pretraining priors. For SFT, we fine-tune each base model on VideoKR-SFT-201K for one epoch. Starting from the resulting SFT checkpoint, we then run GRPO on VideoKR-RL-114K for one epoch. For Qwen3-VL-8B-Instruct, we also conduct Zero-RL training by directly running GRPO on VideoKR-RL-114K for one epoch. We set batch size as 32 for both SFT and GRPO. For the GRPO accuracy reward, we follow prior video reasoning work (Feng et al., 2025a; Wang et al., 2025a) and use ROUGE for open-ended QAs and Exact Match for multiple-choice QAs. The maximum video token number is 4,096, and the maximum number of frames is 128. The training hyperparameters and details are provided in Appendix C.1.

### 5.2. Evaluation Setup

We next describe our evaluation benchmarks and the standardized protocol for fair, reproducible model comparisons.

**Evaluation Benchmarks.** We evaluate models on seven benchmarks grouped into two categories: (1) *General Video Reasoning*, including Video-MME (Fu et al., 2025), MVBench (Li et al., 2024b) and LongVideoBench (Wu et al., 2024), which measure broad video understanding; and (2) *Knowledge-intensive Video Reasoning*, including VideoMMMU (Hu et al., 2025), MMVU (Zhao et al., 2025b),

*Table 3.* Benchmark results across general and knowledge-intensive video reasoning. Models are grouped into (i) *Other Models* and (ii) methods built on Qwen2.5-VL-7B-Instruct or Qwen3-VL-8B-Instruct (with the indicated input Frames). Within each group for (ii), the best score is **bold** and the second-best is underlined.

| Model | Release | Frames | General Video Reasoning | | | | Knowledge-Intensive Video Reasoning | | | | |
|---|---|---|---|---|---|---|---|---|---|---|---|
| | | | Video-MME | MVBench | LongVideoBench | Average | VideoMMMU | MMVU | SciVideoBench | VideoKR-Eval | Average |
| *Closed-source models* | | | | | | | | | | | |
| GPT-5.4 | 2026-03 | 64 | 86.0 | 78.3 | 76.7 | 80.3 | 87.1 | 82.0 | 52.9 | 63.2 | 71.3 |
| Gemini 3 Pro | 2025-11 | 64 | 87.7 | 74.1 | 77.4 | 79.7 | 87.6 | 77.5 | 50.4 | 60.3 | 69.0 |
| Claude Opus 4.5 | 2025-11 | 64 | 81.4 | 67.2 | 67.2 | 71.9 | 84.4 | 77.3 | 48.6 | 56.3 | 66.7 |
| *Other Models* | | | | | | | | | | | |
| Qwen3-VL-32B-Thinking | 2025-10 | 128 | 72.9 | 71.7 | 63.8 | 69.5 | 69.6 | 67.8 | 42.1 | 50.2 | 57.4 |
| Qwen3-VL-32B-Instruct | 2025-10 | 128 | 74.4 | 72.6 | 65.5 | 70.8 | 72.0 | 68.2 | 39.7 | 45.0 | 56.2 |
| Qwen2.5-VL-72B | 2025-02 | 128 | 72.8 | 67.4 | 63.2 | 67.8 | 67.0 | 65.1 | 38.9 | 42.6 | 53.4 |
| InternVL3.5-8B | 2025-08 | 128 | 65.5 | 73.3 | 61.0 | 66.6 | 57.2 | 54.0 | 24.5 | 35.4 | 42.8 |
| LLaVA-OneVision-7B | 2024-08 | 32 | 59.0 | 58.1 | 56.5 | 57.9 | 36.2 | 43.1 | 16.2 | 23.5 | 29.8 |
| LLaVA-NeXT-Video-34B | 2024-07 | 32 | 51.0 | 48.0 | 49.9 | 49.6 | 19.1 | 39.4 | 16.0 | 18.8 | 23.3 |
| LLaVA-NeXT-Video-7B | 2024-07 | 32 | 32.0 | 38.1 | 38.9 | 36.3 | 21.8 | 28.1 | 10.9 | 14.7 | 18.9 |
| *Qwen2.5-VL-7B-Instruct or Qwen3-VL-8B-Instruct as Base Models* | | | | | | | | | | | |
| Qwen2.5-VL-7B-Instruct | 2025-02 | 16 | 57.1 | 65.0 | 55.2 | 59.1 | 48.4 | 52.5 | 23.1 | 31.3 | 38.8 |
| Video-R1 | 2025-03 | 16 | **59.7** | 65.5 | 55.3 | **60.2**$_{+1.1}$ | 51.1 | 53.3 | 26.6 | 28.9 | 40.0$_{+1.2}$ |
| VideoRFT | 2025-05 | 16 | 57.6 | 61.7 | 53.6 | 57.6$_{-1.5}$ | 51.1 | 53.6 | 26.3 | 29.8 | 40.2$_{+1.4}$ |
| **VideoKR (SFT + RL)** | 2026-05 | 16 | 56.6 | **66.6** | **57.0** | 60.1$_{+1.0}$ | **52.6** | **59.2** | **27.3** | **37.7** | **44.2**$_{+5.4}$ |
| Qwen2.5-VL-7B-Instruct | 2025-02 | 128 | 65.1 | 66.3 | 60.9 | 64.1 | 51.1 | 55.7 | 28.1 | 32.7 | 41.9 |
| VideoAuto-R1 | 2026-01 | 128 | **66.8** | **70.2** | 59.7 | **65.6**$_{+1.5}$ | 52.1 | 55.7 | **32.7** | 36.5 | 44.3$_{+2.4}$ |
| **VideoKR (SFT + RL)** | 2026-05 | 128 | 66.4 | 68.9 | **61.3** | 65.5$_{+1.4}$ | **52.2** | **60.5** | 32.5 | **41.2** | **46.6**$_{+4.7}$ |
| Qwen3-VL-8B-Instruct | 2025-10 | 128 | 68.2 | 67.9 | **61.6** | 65.9 | 61.8 | 59.6 | 33.4 | 39.0 | 48.5 |
| OneThinker | 2025-12 | 128 | 65.8 | **69.3** | 61.4 | 65.5$_{-0.4}$ | 62.9 | 61.6 | **33.8** | 38.3 | 49.2$_{+0.7}$ |
| VideoAuto-R1 | 2026-01 | 128 | **68.7** | 68.8 | 58.8 | 65.4$_{-0.5}$ | 63.1 | 59.6 | 32.7 | 43.8 | 49.8$_{+1.3}$ |
| Qwen3-VL-8B-Thinking | 2025-10 | 128 | 67.6 | 68.0 | 60.0 | 65.2$_{-0.7}$ | **64.9** | 60.5 | 33.0 | 41.5 | 50.0$_{+1.5}$ |
| **VideoKR (SFT)** | 2026-05 | 128 | 64.8 | 63.6 | 58.5 | 62.3$_{-3.6}$ | 61.7 | 63.0 | 28.3 | 43.6 | 49.2$_{+0.7}$ |
| **VideoKR (zero RL)** | 2026-05 | 128 | 67.4 | 65.5 | 60.0 | 64.3$_{-1.6}$ | 61.9 | 63.5 | 32.5 | 44.6 | 50.6$_{+2.1}$ |
| **VideoKR (SFT + RL)** | 2026-05 | 128 | 67.8 | 67.0 | 61.5 | 65.4$_{-0.5}$ | 63.0 | **64.8** | 32.8 | **45.3** | **51.5**$_{+3.0}$ |

SciVideoBench (Deng et al., 2025), and VideoKR-Eval, which focus on domain-specific, expert-level reasoning.

**Reproducibility Challenges in Prior Post-Training Work.** We observe substantial cross-paper inconsistencies in reported results, particularly for base models used as post-training starting points. Based on careful follow-up experiments, we attribute these discrepancies primarily to prompt misalignment: base models are sometimes evaluated under prompt conditions that are misaligned with their intended inference mode. For example, Qwen2.5-VL-Instruct is not a "reasoning" model, yet some papers evaluate it with elaborate self-reflection and forced reasoning-trace instructions designed for post-trained 'reasoning' variants.

**Standardizing Evaluation for Fair Model Comparisons.** To ensure fair and reproducible comparisons, for each model, we use the official prompt released by the original paper whenever available; otherwise, we adopt the default prompt templates from LMMs-Eval (Li et al., 2024a). For input frames, we follow the model-recommended inference configuration whenever it is specified by the model release. We run each model three times with independent sampling and report the mean. All evaluations are performed using the LMMs-Eval framework (Li et al., 2024a). Full evaluation details (*e.g.,* model prompts and inference parameters) are provided in Appendix C.2.

## 6. Experiment Results and Analysis

We next discuss our main findings and ablation analysis.

### 6.1. Main Results

Table 3 presents the main experimental results. Post-training on VideoKR consistently improves Qwen2.5-VL-7B-Instruct across all evaluated benchmarks, and yields clear gains for Qwen3-VL-8B-Instruct especially on knowledge-intensive video reasoning benchmarks. The gains are most pronounced on knowledge-intensive tasks: SFT followed by RL on VideoKR raises the knowledge-

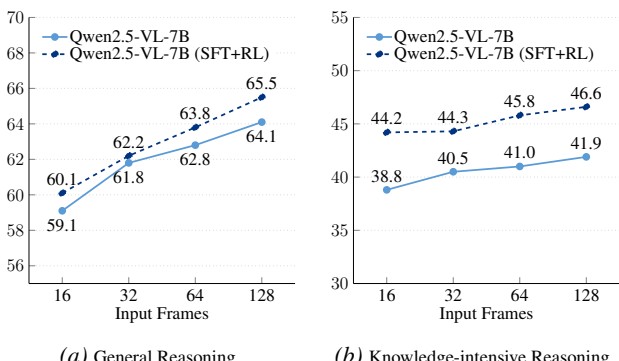

*(a)* General Reasoning    *(b)* Knowledge-intensive Reasoning

*Figure 3.* Inference-time frame scaling results on general and knowledge-intensive video reasoning benchmarks. The figure shows category-wise average accuracies for Qwen2.5-VL-7B-Instruct and its VideoKR post-trained variant (SFT+RL) under different input frame budgets. Appendix D.1 provides the full per-benchmark results for post-trained Qwen2.5-VL-7B-Instruct and Qwen3-VL-8B-Instruct models.

intensive average of Qwen2.5-VL-7B from 41.9 to 46.6 (+4.7) and of Qwen3-VL-8B from 48.5 to 51.5 (+3.0), with the largest per-dataset improvements on MMVU and VideoKR-Eval (*e.g.,* +4.8 and +8.5 points for Qwen2.5-VL-7B). Notably, the post-trained Qwen3-VL-8B attains the best knowledge-intensive average among 7/8B-scale models (51.5, versus 50.0 for the strongest competing model, Qwen3-VL-8B-Thinking). These results underscore the value of our training data, which tightly integrates domain knowledge, visual grounding, and structured reasoning. Starting from an SFT-initialized model, subsequent RL training consistently delivers higher performance than the SFT-only baseline, highlighting that combining SFT with RL is important for fully leveraging the strengths of VideoKR data. Moreover, the RL-only variant also generally outperforms the SFT-only model, indicating that RL alone can induce stronger generalizable reasoning abilities.

## 6.2. Case Study

To better understand how post-training on VideoKR changes model behavior, we randomly sample 100 examples from VideoKR-Eval and compare outputs from different models. Our case study shows that the post-trained Qwen3-VL-8B model can integrate visual evidence with relevant domain knowledge to perform complex reasoning. It also exhibits "aha-moment" reasoning patterns indicative of deeper understanding. Examples are shown in Appendix D.3.

## 6.3. Analysis of Inference-Time Frame Scaling

To analyze how input frame count affects performance, we evaluate our post-trained model, which was trained with 128 frames, under varying numbers of frames at inference. Specifically, we test 16, 32, 64, and 128 frames while keep-

ing all other inference settings fixed. As shown in Figure 3, increasing the number of input frames consistently improves performance for both the base model and our post-trained model. For example, on general video reasoning benchmarks, Qwen2.5-VL-7B (SFT+RL) improves from 60.1% at 16 frames to 65.5% at 128 frames. On knowledge-intensive video reasoning benchmarks, it further improves from 44.2% to 46.6%. These results suggest that our model benefits from richer visual and temporal evidence at inference time, and that the gains from VideoKR remain consistent across different frame budgets.

In the following subsections, we conduct controlled SFT and RL ablations on VideoKR to provide insights for future work. Unless otherwise specified, all experiments use Qwen2.5-VL-7B-Instruct as the base model and are trained and evaluated with 128 input frames.

## 6.4. Ablations on VideoKR-SFT-201K

Using VideoKR-SFT-201K, we ablate two design choices, the skill composition of the training examples and the use of CoT supervision, to quantify their individual contributions.

**Skill-Oriented Data Composition.** To assess the contribution of each skill component, we fine-tune models on cumulative subsets of our skill-oriented data. Specifically, we construct three 80K-example variants from VideoKR-SFT-201K: (1) VIDR only; (2) a balanced mixture of VIDR and KNOWVID (1:1); and (3) a randomly sampled subset from the full VideoKR-SFT-201K (*i.e.,* VIDR +KNOWVID +KNOWVIDR, we reuse examples in the previous ablation). We fine-tune Qwen2.5-VL-7B-Instruct on each variant for one epoch with a batch size of 16. We observe that incorporating all three skill components yields the best knowledge-intensive performance: training on VIDR alone obtains 41.4% on knowledge-intensive benchmarks. Adding Knowledge-Enhanced Perception (VIDR +KNOWVID) gives 41.3%, while incorporating Knowledge-Intensive Reasoning (VIDR +KNOWVID +KNOWVIDR) further improves performance to 42.4%. The same trend holds on VideoKR-Eval, where accuracy rises monotonically from 35.3 (VIDR) to 35.9 (VIDR +KNOWVID) to 36.8 (VIDR +KNOWVID +KNOWVIDR). These results indicate that combining domain knowledge and complex reasoning supervision is crucial.

**CoT vs. Direct Output.** To validate the necessity of explicit CoT rationales in SFT, we randomly sample 80K examples from VideoKR-SFT-201K and create two variants: one with CoT rationales and one without. We fine-tune Qwen2.5-VL-7B-Instruct on each variant for one epoch with a batch size of 16. As illustrated in Table 4, the CoT-trained model improves over the Direct Output baseline on knowledge-intensive reasoning, lifting the average from 39.4% to 42.4% (a 3.0-point gain), underscoring the

*Table 4.* Ablation studies on post-training data. All experiments use Qwen2.5-VL-7B-Instruct as the base model, with 128 input frames. The complete results are provided in Appendix D.2.

| Ablation Setting | General | Knowledge-Intensive | |
|---|---|---|---|
| | Average | VideoKR-Eval | Average |
| Qwen2.5-VL-7B-Instruct | 64.1 | 32.7 | 41.9 |
| *Skill-Oriented Data Composition (SFT, 80K examples, one epoch)* | | | |
| VIDR | $58.0_{-6.1}$ | $35.3_{+2.6}$ | $41.4_{-0.5}$ |
| VIDR + KNOWVID | $58.4_{-5.7}$ | $35.9_{+3.2}$ | $41.3_{-0.6}$ |
| VIDR + KNOWVID + KNOWVIDR | $58.3_{-5.8}$ | $36.8_{+4.1}$ | $42.4_{+0.5}$ |
| *CoT Supervision Format (SFT, 80K examples, one epoch)* | | | |
| Direct Output | $61.4_{-2.7}$ | $35.9_{+3.2}$ | $39.4_{-2.5}$ |
| Chain-of-Thought | $58.3_{-5.8}$ | $36.8_{+4.1}$ | $42.4_{+0.5}$ |
| *Comparison with Other SFT Corpora (SFT, 80K examples, one epoch)* | | | |
| Video-R1-CoT-165k | $57.3_{-6.8}$ | $27.5_{-5.2}$ | $36.2_{-5.7}$ |
| OneThinker-SFT-340k | $60.5_{-3.6}$ | $29.7_{-3.0}$ | $38.3_{-3.6}$ |
| VideoRFT-CoT-102K | $59.6_{-4.5}$ | $32.1_{-0.6}$ | $38.4_{-3.5}$ |
| VideoKR-SFT-201K (Ours) | $58.3_{-5.8}$ | $36.8_{+4.1}$ | $42.4_{+0.5}$ |
| *Comparison with Other RL Corpora (GRPO, 50K examples, one epoch)* | | | |
| Video-R1-260k | $63.7_{-0.4}$ | $33.1_{+0.4}$ | $41.6_{-0.3}$ |
| OneThinker-600k | $60.3_{-3.8}$ | $33.2_{+0.5}$ | $42.3_{+0.4}$ |
| VideoRFT-RL-310K | $63.8_{-0.3}$ | $33.5_{+0.8}$ | $42.3_{+0.4}$ |
| VideoAuto-R1-83K | $62.8_{-1.3}$ | $33.3_{+0.6}$ | $42.7_{+0.8}$ |
| VideoKR-RL-114K (Ours) | $61.7_{-2.4}$ | $34.5_{+1.8}$ | $43.0_{+1.1}$ |

*Table 5.* Accuracy of Qwen2.5/3-VL models on 3,000 randomly sampled QA examples from various post-training corpora.

| Model | Video-R1 | VideoRFT | OneThinker | VidAuto-R1 | VideoKR |
|---|---|---|---|---|---|
| 2.5-VL-7B-Inst. | 55.3 | 47.8 | 45.8 | 57.1 | 39.2 |
| 3-VL-8B-Inst. | 57.1 | 51.1 | 49.1 | 54.5 | 42.3 |
| 3-VL-8B-Think | 59.0 | 52.3 | 49.3 | 54.3 | 43.5 |

importance of high-quality CoT supervision for advanced knowledge-intensive video reasoning.

### 6.5. VideoKR vs Prior Post-Training Corpus

We next conduct a comprehensive comparison of VideoKR with prior open-source post-training corpora, under both **SFT** and **zero-RL** settings. For SFT, we randomly sample 80K examples from each SFT corpus and fine-tune Qwen2.5-VL-7B-Instruct on each variant for one epoch with a batch size of 16. For zero-RL, we randomly sample 50K QA examples from each RL corpus and train Qwen2.5-VL-7B-Instruct on each variant for one epoch with a batch size of 16; we exclude non-QA tasks since they require task-specific reward functions that cannot be unified under our standard GRPO training pipeline.

**Main Findings.** As illustrated in Table 4, under SFT, the model trained on the VideoKR-SFT subset reaches a 42.4 knowledge-intensive average and is the only corpus to surpass the base model (41.9), whereas prior corpora such as Video-R1 and VideoRFT lower it to 36.2 and 38.4, respectively. Under zero-RL, VideoKR with VideoKR-RL-114K achieves the strongest gain (43.0, +1.1 over the base model, ahead of the next-best VideoAuto-R1 at 42.7), indicating that high-quality data is key to maximizing post-training benefits for advanced video reasoning.

**Training-Data Difficulty Analysis.** To diagnose why prior corpora provide limited improvements relative to VideoKR, we analyze training-data difficulty with respect to the base models. Concretely, we randomly sample 3,000

video QA examples from each corpus and measure the zero-shot accuracy of Qwen2.5-VL-7B and Qwen3-VL-8B in the 128-frame setting. As shown in Table 5, all evaluated models attain high accuracy on prior corpora (*e.g.,* Qwen3-VL-8B scores between 49.1% and 57.1%), suggesting these datasets are effectively saturated for current frontier base models and thus offer weak learning signals. In contrast, accuracy on VideoKR remains lower (42.3% for the same model), indicating a more challenging distribution that better supports continued capability gains during post-training.

## 7. Conclusion

This work offers a corpus-centric perspective on post-training foundation models for advanced video reasoning. Instead of viewing visual perception, domain knowledge, and advanced reasoning as loosely linked elements, we show that integrating structured domain concepts with visually grounded examples yields stronger reasoning performance, without relying on sophisticated RL reward engineering. Extensive experiments and analyses confirm that post-training on the VideoKR dataset produces strong improvements, especially on knowledge-intensive video reasoning.

## Impact Statement

All videos in VideoKR are licensed under CC license, which enables reusers to distribute, remix, adapt, and build upon the material in any medium or format, so long as attribution is given to the creator. By restricting our dataset to CC-licensed videos, we ensure legal reusability and clear provenance, a critical aspect that has been ambiguous in prior training corpora. For human expert annotation and validation, we compensated annotators at an average rate of $13 USD per hour, which exceeds the prevailing hourly rates for comparable local work. All expert annotators provided informed consent to participate and explicitly authorized the public release and redistribution of their annotations as part of the resulting dataset and accompanying materials. The VideoKR and VideoKR-Eval data construction incurs approximately 70.4K US dollars in model inference costs.

## Acknowledgments

We thank Dr. Yu Rong for valuable suggestions on the post-training experiment design and paper writeup improvement.

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

# A. VideoKR Data Construction

## A.1. Domain Knowledge Bank Construction

*Table 6.* Complete subject list by major disciplines. Columns list subfields under each of the four major disciplines.

| Natural Sciences (20) | Engineering (20) | Healthcare (18) | Humanities & Social Sciences (24) |
|---|---|---|---|
| Chemistry | Electrical Engineering | Medicine | History |
| Physics | Computer Science | Public Health | Psychology |
| Biology | Materials Science & Engineering | Pharmacy | Arts |
| Astrophysics & Astronomy | Mechanical Engineering | Dentistry | Education |
| Earth Science | Civil & Environmental Engineering | Nursing | Sociology |
| Mathematics | Chemical Engineering | Biomedical Sciences | Philosophy |
| Statistics & Data Science | Biomedical Engineering | Medical Laboratory Science | Linguistics |
| Environmental Science | Aerospace Engineering | Nutrition & Dietetics | Anthropology |
| Ecology & Evolutionary Biology | Industrial & Operations Research | Physiotherapy | Archaeology |
| Microbiology & Immunology | Systems Engineering | Occupational Therapy | Political Science |
| Biochemistry & Molecular Biology | Nuclear Engineering | Speech & Language Therapy | International Relations |
| Biophysics | Energy Engineering | Radiography / Imaging Sciences | Economics |
| Neuroscience | Mechatronics & Robotics | Health / Biomedical Informatics | Law |
| Genetics & Genomics | Software Engineering | Epidemiology & Biostatistics | Geography |
| Cell & Developmental Biology | AI Engineering | Global Health | Communication & Media Studies |
| Marine Science | Computer Engineering | Veterinary Medicine | Literature & Comparative Literature |
| Atmospheric Science | Communications Engineering | Optometry / Vision Science | Modern Languages & Cultures |
| Geology & Geophysics | Control & Automation | Health Policy & Management | Theology & Religious Studies |
| Paleontology | Structural Engineering | | Business Administration |
| Scientific Computing | Geotechnical Engineering | | Finance |
| | | | Accounting |
| | | | Architecture & Urban Planning |
| | | | Public Policy & Administration |
| | | | Gender & Sexuality Studies |

Based on a manual review of undergraduate curricula from leading universities worldwide, we identified 82 representative subjects spanning four major disciplines. Table 6 organizes these subjects across Natural Sciences, Healthcare, Humanities and Social Sciences, and Engineering, forming the top-level index of our four-layer knowledge base of subject, course, lecture, and knowledge point, and enabling broad cross-domain coverage and balanced sampling.

## A.2. Annotator Information

*Table 7.* Biographies of 34 annotators involved in the VideoKR construction pipeline. The table details their participation in: **Know. Bank** (Domain Knowledge Bank Construction), **Seed Ex.** (Seed Example Curation), **Model Val.** (Human-Validated Model Selection), **Quality** (Manual Quality Assessment), and **Eval Bench.** (VideoKR-Eval Construction).

| ID | Year | Major | Assigned Discipline | Annotation Tasks | | | | |
|---|---|---|---|---|---|---|---|---|
| | | | | Know. Bank | Seed Ex. | Model Val. | Quality | Eval Bench. |
| 1 | 3rd yr PhD | Electrical Eng. | Engineering | ✓ | ✓ | ✓ | ✓ | ✓ |
| 2 | 3rd yr PhD | Mechanical Eng. | Engineering | ✓ | ✓ | ✓ | ✓ | ✓ |
| 3 | 1st yr PhD | Computer Science | Engineering | | ✓ | ✓ | ✓ | ✓ |
| 4 | 2nd yr PhD | Civil Engineering | Engineering | ✓ | ✓ | ✓ | | |
| 5 | 2nd yr Master | Chemical Eng. | Engineering | ✓ | ✓ | | | |
| 6 | 1st yr Master | Materials Science | Engineering | | ✓ | ✓ | | ✓ |
| 7 | 1st yr Master | Aerospace Eng. | Engineering | | ✓ | ✓ | | |
| 8 | 2nd yr Master | Biomedical Eng. | Engineering | ✓ | ✓ | | | ✓ |
| 9 | 2nd yr Master | Software Eng. | Engineering | | ✓ | | | |
| 10 | 1st yr Master | Electronic Eng. | Engineering | | ✓ | ✓ | | |
| 11 | 1st yr Master | Industrial Eng. | Engineering | ✓ | ✓ | | | |
| 12 | 3rd yr PhD | Physics | Natural Sciences | ✓ | ✓ | ✓ | ✓ | ✓ |
| 13 | 2nd yr PhD | Chemistry | Natural Sciences | ✓ | ✓ | ✓ | ✓ | ✓ |
| 14 | 2nd yr PhD | Biology | Natural Sciences | | ✓ | ✓ | ✓ | ✓ |
| 15 | 1st yr PhD | Mathematics | Natural Sciences | ✓ | ✓ | ✓ | | |
| 16 | 2nd yr Master | Statistics | Natural Sciences | ✓ | ✓ | | | ✓ |
| 17 | 1st yr Master | Earth Science | Natural Sciences | | ✓ | ✓ | | |
| 18 | 1st yr Master | Astrophysics | Natural Sciences | | ✓ | ✓ | | |
| 19 | 1st yr Master | Environmental Sci. | Natural Sciences | ✓ | ✓ | | | |
| 20 | 2nd yr Master | Geology | Natural Sciences | ✓ | ✓ | | | |
| 21 | 1st yr Master | Ecology | Natural Sciences | | ✓ | ✓ | | |
| 22 | 2nd yr PhD | Economics | Humanities & Social Sciences | ✓ | ✓ | ✓ | ✓ | ✓ |
| 23 | 2nd yr PhD | Psychology | Humanities & Social Sciences | ✓ | ✓ | ✓ | ✓ | ✓ |
| 24 | 3rd yr PhD | Sociology | Humanities & Social Sciences | | ✓ | | | ✓ |
| 25 | 1st yr Master | Political Science | Humanities & Social Sciences | ✓ | ✓ | | | |
| 26 | 1st yr Master | Philosophy | Humanities & Social Sciences | ✓ | ✓ | ✓ | | ✓ |
| 27 | 1st yr Master | History | Humanities & Social Sciences | ✓ | ✓ | ✓ | | |
| 28 | 1st yr Master | Law | Humanities & Social Sciences | ✓ | ✓ | | | ✓ |
| 29 | 2nd yr Master | Linguistics | Humanities & Social Sciences | | ✓ | | | ✓ |
| 30 | 1st yr Master | Education | Humanities & Social Sciences | | ✓ | ✓ | | |
| 31 | 4th yr PhD | Public Health | Healthcare | ✓ | ✓ | ✓ | ✓ | ✓ |
| 32 | 2nd yr PhD | Clinical Medicine | Healthcare | ✓ | ✓ | ✓ | ✓ | ✓ |
| 33 | 2nd yr PhD | Dentistry | Healthcare | | ✓ | | | |
| 34 | 2nd yr Master | Pharmacy | Healthcare | ✓ | ✓ | | | ✓ |

## A.3. VideoKR Data Example

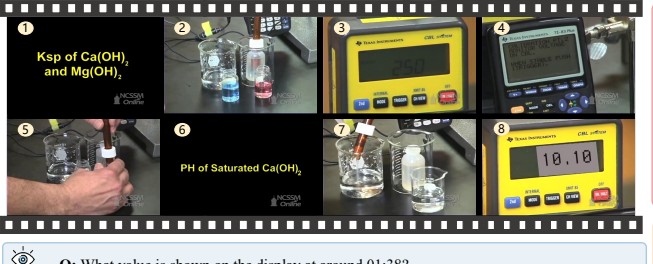

**Q:** Which tested sample shows formation of an insoluble solid during measurement?

**Answer:** magnesium hydroxide sample

**Reasoning:** Let me analyze the video content around the relevant measurement mark. Looking at the sequence: At around 01:53, the text "PH of Saturated $Ca(OH)_2$" appears,(...abbreviated...) The question asks about which sample forms an insoluble solid during measurement. However, I need to consider the chemistry involved:(...abbreviated...) The video is demonstrating the difference in solubility between these two hydroxides, and the formation of insoluble solid would be most notable with the magnesium hydroxide sample due to its much lower Ksp value.

**Q:** A saturated solution of $Mg(OH)_2$ exhibits a pH value measured in the experiment. Calculate the Ksp of $Mg(OH)_2$ using the observed pH.

A. $1.0 \times 10^{-12}$      B. $5.6 \times 10^{-12}$
C. $1.8 \times 10^{-11}$      D. $4.0 \times 10^{-12}$

**Answer:** A

**Reasoning:** To calculate the Ksp of $Mg(OH)_2$, first extract the saturated solution's pH = 10.10 from the video(...abbreviated...). Calculate pOH = 14.00 - 10.10 = 3.90, then $[OH^-] = 10^{-3.90} \approx 1.2589 \times 10^{-4}$ M(...abbreviated...). From $Mg(OH)_2 \rightleftharpoons Mg^{2+} + 2OH^-$, $[Mg^{2+}] = [OH^-]/2 \approx 6.2945 \times 10^{-5}$ M(...abbreviated...). Substitute into Ksp = $[Mg^{2+}][OH^-]^2$: Ksp = $(6.2945 \times 10^{-5})(1.2589 \times 10^{-4})^2 \approx 9.98 \times 10^{-13}$, rounding to $1.0 \times 10^{-12}$.

**Q:** What value is shown on the display at around 01:38?

**Answer:** −6.419

**Reasoning:** The user is asking about what value is displayed at approximately 1 minute and 38 seconds (98 seconds) into the video. Looking at the timestamp at around 98 seconds, I can see a Texas Instruments TI-83 Plus calculator display. The screen shows statistical calculation results, including: - "INTERCEPT= -6.419" - "SLOPE=11.824" The question asks for "the value" shown at this time, and given the correct answer provided is -6.419, this corresponds to the intercept value displayed on the calculator screen.

*Figure 4.* A VideoKR-SFT-201K example from the natural science domain. The reasoning process is presented in a concise and abbreviated form to improve readability.

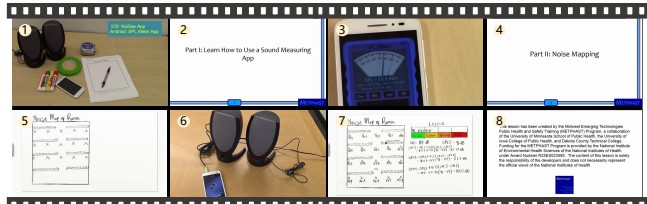

**Q:** At the moment the handheld display shows its highest reading, which category applies to that spot on the diagram?

**Answer:** Red

**Reasoning:** Looking at the video timeline, I need to identify when the handheld SPL meter shows its highest reading and then determine which category that corresponds to on the noise map.(...abbreviated...)The highest reading on the handheld SPL meter is 79.1 dBA, recorded at around 101 and 109 seconds.(...abbreviated...)This peak reading corresponds to the red category on the noise map's Green-Yellow-Orange-Red color legend, which is reserved for the loudest measured noise levels in the room.

**Q:** During the SPL meter demonstration, at the moment when the current SPL value is displayed as 77.2 dBA, calculate the ratio of (maximum value - average value) to (average value - minimum value). Round to two decimal places.

A. 1.85      B. 2.03
C. 2.15      D. 3.03

**Answer:** B

**Reasoning:** Extract the three key values of the SPL meter shown in the video when the current SPL value is 77.2 dBA(...abbreviated...), the maximum value is 79.5 dBA, the minimum value is 76.5 dBA and the average value is 77.49 dBA. First calculate numerator: 79.5 - 77.49 = 2.01, then calculate the denominator: 77.49 - 76.5 = 0.99(...abbreviated...). Divide the two results to get the ratio: 2.01 ÷ 0.99 ≈ 2.0303, and round it to two decimal places to get 2.03.

**Q:** What device is physically connected to the phone before measurements begin?

**Answer:** External speakers

**Reasoning:** I need to examine the video carefully to identify what device is physically connected to the phone before the measurement phase begins.(...abbreviated...)Then at around 194.6 seconds onwards, I can observe a table setup showing a phone positioned between two black external speakers. The speakers appear to be wired and connected to the phone with visible cables. This setup is shown clearly before the person begins walking around the room taking measurements (which occurs around 225.7 seconds).(...abbreviated...)The external speakers are part of the controlled setup for the noise mapping exercise - they likely serve as the sound source that participants are trying to locate and map using the sound measuring app on their phone.

*Figure 5.* A VideoKR-SFT-201K example from the healthcare domain. The reasoning process is presented in a concise and abbreviated form to improve readability.

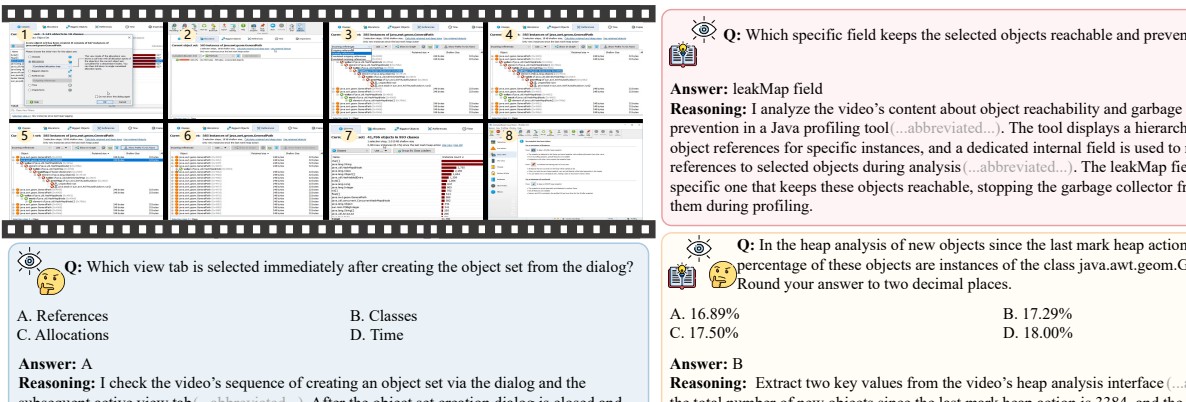

**Q:** Which specific field keeps the selected objects reachable and prevents reclamation?

**Answer:** leakMap field
**Reasoning:** I analyze the video's content about object reachability and garbage collection prevention in a Java profiling tool(...abbreviated...). The tool displays a hierarchical tree of object references for specific instances, and a dedicated internal field is used to maintain references to selected objects during analysis(...abbreviated...). The leakMap field is the specific one that keeps these objects reachable, stopping the garbage collector from reclaiming them during profiling.

**Q:** Which view tab is selected immediately after creating the object set from the dialog?

A. References
B. Classes
C. Allocations
D. Time

**Answer:** A
**Reasoning:** I check the video's sequence of creating an object set via the dialog and the subsequent active view tab(...abbreviated...). After the object set creation dialog is closed and the new object set is generated, the interface displays content with "Incoming references" and a corresponding tree structure(...abbreviated...). This visual confirms the References tab is the active view immediately after the action, so the answer is A.

**Q:** In the heap analysis of new objects since the last mark heap action, what percentage of these objects are instances of the class java.awt.geom.GeneralPath? Round your answer to two decimal places.

A. 16.89%
B. 17.29%
C. 17.50%
D. 18.00%

**Answer:** B
**Reasoning:** Extract two key values from the video's heap analysis interface(...abbreviated...): the total number of new objects since the last mark heap action is 3384, and the instance count of java.awt.geom.GeneralPath is 585. Calculate the percentage with the formula (target class instances / total new objects) × 100(...abbreviated...). Substitute the values: (585 ÷ 3384) × 100 ≈ 17.2872%, round to two decimal places to get 17.29%, which corresponds to option B.

*Figure 6.* A VideoKR-SFT-201K example from the engineering domain. The reasoning process is presented in a concise and abbreviated form to improve readability.

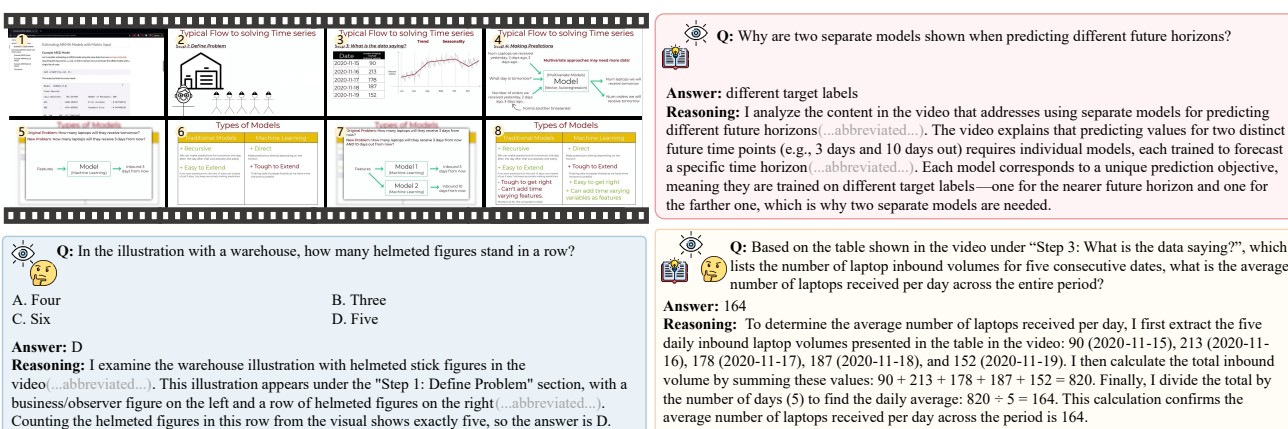

**Q:** Why are two separate models shown when predicting different future horizons?

**Answer:** different target labels
**Reasoning:** I analyze the content in the video that addresses using separate models for predicting different future horizons(...abbreviated...). The video explains that predicting values for two distinct future time points (e.g., 3 days and 10 days out) requires individual models, each trained to forecast a specific time horizon(...abbreviated...). Each model corresponds to a unique prediction objective, meaning they are trained on different target labels—one for the nearer future horizon and one for the farther one, which is why two separate models are needed.

**Q:** In the illustration with a warehouse, how many helmeted figures stand in a row?

A. Four
B. Three
C. Six
D. Five

**Answer:** D
**Reasoning:** I examine the warehouse illustration with helmeted stick figures in the video(...abbreviated...). This illustration appears under the "Step 1: Define Problem" section, with a business/observer figure on the left and a row of helmeted figures on the right(...abbreviated...). Counting the helmeted figures in this row from the visual shows exactly five, so the answer is D.

**Q:** Based on the table shown in the video under "Step 3: What is the data saying?", which lists the number of laptop inbound volumes for five consecutive dates, what is the average number of laptops received per day across the entire period?

**Answer:** 164
**Reasoning:** To determine the average number of laptops received per day, I first extract the five daily inbound laptop volumes presented in the table in the video: 90 (2020-11-15), 213 (2020-11-16), 178 (2020-11-17), 187 (2020-11-18), and 152 (2020-11-19). I then calculate the total inbound volume by summing these values: 90 + 213 + 178 + 187 + 152 = 820. Finally, I divide the total by the number of days (5) to find the daily average: 820 ÷ 5 = 164. This calculation confirms the average number of laptops received per day across the period is 164.

*Figure 7.* A VideoKR-SFT-201K example from the engineering domain. The reasoning process is presented in a concise and abbreviated form to improve readability.

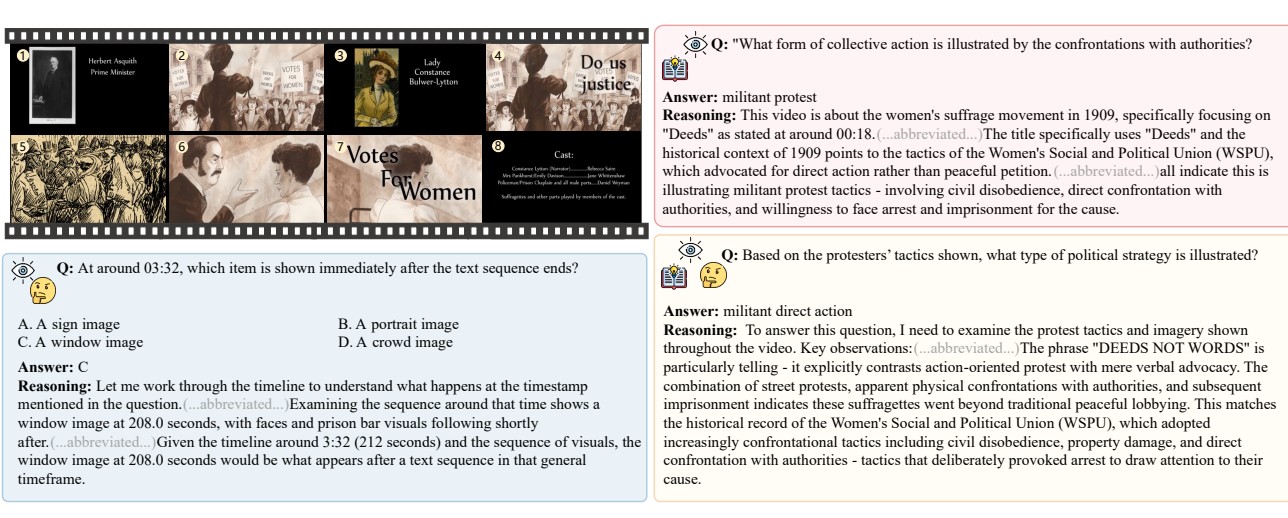

**Q:** "What form of collective action is illustrated by the confrontations with authorities?

**Answer:** militant protest
**Reasoning:** This video is about the women's suffrage movement in 1909, specifically focusing on "Deeds" as stated at around 00:18.(...abbreviated...)The title specifically uses "Deeds" and the historical context of 1909 points to the tactics of the Women's Social and Political Union (WSPU), which advocated for direct action rather than peaceful petition.(...abbreviated...)all indicate this is illustrating militant protest tactics - involving civil disobedience, direct confrontation with authorities, and willingness to face arrest and imprisonment for the cause.

**Q:** At around 03:32, which item is shown immediately after the text sequence ends?

A. A sign image
B. A portrait image
C. A window image
D. A crowd image

**Answer:** C
**Reasoning:** Let me work through the timeline to understand what happens at the timestamp mentioned in the question.(...abbreviated...)Examining the sequence around that time shows a window image at 208.0 seconds, with faces and prison bar visuals following shortly after.(...abbreviated...)Given the timeline around 3:32 (212 seconds) and the sequence of visuals, the window image at 208.0 seconds would be what appears after a text sequence in that general timeframe.

**Q:** Based on the protesters' tactics shown, what type of political strategy is illustrated?

**Answer:** militant direct action
**Reasoning:** To answer this question, I need to examine the protest tactics and imagery shown throughout the video. Key observations:(...abbreviated...)The phrase "DEEDS NOT WORDS" is particularly telling - it explicitly contrasts action-oriented protest with mere verbal advocacy. The combination of street protests, apparent physical confrontations with authorities, and subsequent imprisonment indicates these suffragettes went beyond traditional peaceful lobbying. This matches the historical record of the Women's Social and Political Union (WSPU), which adopted increasingly confrontational tactics including civil disobedience, property damage, and direct confrontation with authorities - tactics that deliberately provoked arrest to draw attention to their cause.

*Figure 8.* A VideoKR-SFT-201K example from the humanities and social science domain. The reasoning process is presented in a concise and abbreviated form to improve readability.

## A.4. Human-Validated Model Selection Protocol

To ensure that the VideoKR corpus is constructed with the highest possible quality while maximizing data diversity, we implemented a rigorous, human-in-the-loop qualification protocol for all foundation models used in our pipeline. Instead of relying on a single model (which risks imprinting specific model biases), we maintain a dynamic pool of eligible models for each synthesis stage. This section details the qualification methodology and the resulting model assignments.

We evaluate seven frontier models for potential inclusion in each pipeline step: GPT-5.2, GPT-5-mini, Claude-4.5-Sonnet, Gemini-3-Flash, DeepSeek-V3.2, Qwen3-VL-235B-A22B, and GLM-4.6V. For every pipeline stage defined in §3.1–§3.4, we conducted a controlled pilot study by sampling 100 representative input instances. Domain experts evaluated the model outputs against strict criteria, distinguishing between hard compliance failures (*e.g.,* JSON format violations) and soft content failures (*e.g.,* hallucinations or weak reasoning). A model was deemed eligible for a specific stage only if its total error rate $\leq 3\%$. Table 8 shows the human-validated models for each pipeline step.

*Table 8.* The eligible models at each VideoKR construction pipeline step. "-" indicates the model is not optimized for the modality required at that step; "✗" indicates the model fails human validation for that step.

| Pipeline Stage | Modality | GPT-5.2 | GPT-5-mini | Claude-4.5-Sonnet | Gemini-3-Flash | DeepSeek-V3.2 | Qwen3-VL-235B-A22B | GLM-4.6V |
|---|---|---|---|---|---|---|---|---|
| *§3.1 Domain Knowledge Bank Construction* | | | | | | | | |
| Lecture → Knowledge Point | Text | ✓ | ✓ | ✓ | ✓ | ✓ | – | – |
| *§3.2 Knowledge-Driven Video Collection* | | | | | | | | |
| Scenario Generation | Text | ✓ | ✓ | ✓ | ✓ | ✗ | – | – |
| Search Keyword Generation | Text | ✓ | ✓ | ✓ | ✓ | ✓ | – | – |
| Metadata Relevance Judge | Text | ✓ | ✗ | ✓ | ✗ | ✗ | – | – |
| Visual Relevance Judge | Vision | ✓ | ✗ | ✗ | ✓ | – | ✗ | ✗ |
| *§3.3 Skill-Oriented Example Generation* | | | | | | | | |
| Example Generation | Vision | ✓ | ✗ | ✓ | ✓ | – | ✗ | ✗ |
| CoT Rationale Validation | Vision | ✓ | ✗ | ✓ | ✓ | – | ✗ | ✗ |

## A.5. Data Contamination Mitigation

For *Near-Duplicate Video Filtering*, we uniformly sample both benchmark and VideoKR videos at 1 fps using `ffmpeg` and compute 64-bit perceptual hashes per frame. We partition each training video into overlapping 20-second windows with a 1-second stride and build an index over window-level hash sequences to enable scalable retrieval. For each benchmark video, we retrieve the top-10 candidate training windows and verify them by aligned-frame Hamming distance; we flag an overlap when the best 20-second window has at least $70\%$ of frames with distance $\leq 30$, and remove any matched training video.

# B. VideoKR-Eval Benchmark

## B.1. Detailed Statistics of VideoKR-Eval

As shown in Table 9, we construct VideoKR-Eval from three source benchmarks: VideoMMMU, MMVU, and SciVideoBench. We first perform multi-model single-frame probing and retain only original examples that are judged to require continuous video understanding by all three probing models. This yields 1,254 retained original examples. For the remaining filtered videos, domain experts re-annotate new visually grounded QA examples, contributing 746 additional examples. The final benchmark comprises 2,000 high-quality examples designed to require genuine video-level understanding and knowledge-intensive reasoning.

*Table 9.* Detailed statistics for the VideoKR-Eval benchmark construction. We retain original examples that are judged to require continuous video understanding by all three single-frame probing models, and add expert-reannotated examples for the filtered videos.

| Source Benchmark | Candidate Count | VideoKR-Eval Composition | | | Final Count |
|---|---|---|---|---|---|
| | | Filtered | Retained Original | Expert-Reannotated | |
| MMVU | 1,000 | 639 | 361 | 398 | 759 |
| VideoMMMU | 900 | 560 | 340 | 241 | 581 |
| SciVideoBench | 1,000 | 447 | 553 | 107 | 660 |
| **Total** | **2,900** | **1,646** | **1,254** | **746** | **2,000** |

# C. Experiment Setup

## C.1. Post-Training Details

**GRPO Reward Design.** We employ GRPO (Shao et al., 2024) as our reinforcement learning algorithm. Following the standard RLVR-style reward formulation, the total reward is defined as $R = 0.1 \cdot R_f + 0.9 \cdot R_a$, where $R_f$ and $R_a$ denote the *format* and *accuracy* rewards, respectively. Specifically, $R_f$ is set to $1.0$ if the model output strictly satisfies the required format: `<think>...</think><answer>...</answer>`. For the accuracy reward $R_a$, we adopt the ROUGE metric for open-ended QA, while employing Exact Match (EM) for multiple-choice tasks.

**Training Details.** We train all models on up to 8 NVIDIA A800 GPUs (80 GB). For SFT, we use a learning rate of $1 \times 10^{-5}$, while for RL we use a learning rate of $5 \times 10^{-6}$. Both stages are optimized with AdamW, and the maximum response length is set to 2,048 tokens. For GRPO rollout generation, we set the rollout size $G$ to 8 and use a temperature of 1.0 to encourage exploration. The KL penalty coefficient $\beta$ is set to 0.01. Supervised fine-tuning is implemented with LLaMA-Factory (Zheng et al., 2024), while reinforcement learning is implemented with verl (Sheng et al., 2024).

## C.2. Evaluation Setup

To ensure fair and reproducible comparisons, we standardize the inference configuration for all evaluations by setting the temperature to $0.1$. The maximum response token is set to 8,192 tokens.

---

**All Benchmarks – For VideoKR and OneThinker**

{Question}
Please answer this question based on the visual content.
Provide your thinking process between the `<think>` and `</think>` tags, and then give your final answer between the `<answer>` and `</answer>` tags.
At the end, you must output the final answer in the format:
`<answer><your answer here></answer>`

---

**All Benchmarks – For VideoAuto-R1**

**Video-MME**
Select the best answer to the following multiple-choice question based on the video. Respond with only the letter (A, B, C, or D) of the correct option
{question}
Put your final answer in \\boxed{}.

- - - - - - - - - - - - - - - - - - - - - - - - - - - - - - - - - - - - - - - - - - - - - - -

**Other benchmarks**
{question}
Put your final answer in \\boxed{}.

---

**All Benchmarks – For Video-R1 and VideoRFT**

{Question}
Please think about this question as if you were a human pondering deeply.
Engage in an internal dialogue using expressions such as 'let me think', 'wait', 'Hmm', 'oh, I see', 'let's break it down', etc, or other natural language thought expressions.
It's encouraged to include self-reflection or verification in the reasoning process.
Provide your detailed reasoning between the `<think>` `</think>` tags, and then give your final answer between the `<answer>` `</answer>` tags.
**For multiple-choice:** Please provide only the single option letter (e.g., A, B, C, D, etc.) within the `<answer>` `</answer>` tags.
**For open-ended:** Please provide your text answer within the `<answer>` `</answer>` tags.

Video-MME | MVBench – For Qwen series instruct models

Select the best answer to the following multiple-choice question based on the video. Respond with only the letter (A, B, C, or D) of the correct option.
Question: {question} Possible answer choices:
{options}
The best answer is:

Video-MME | MVBench – For Qwen series thinking models

Select the best answer to the following multiple-choice question based on the video. Respond with only the letter (A, B, C, or D) of the correct option.
Question: {question} Possible answer choices:
{options}
Please reason step-by-step, identify relevant visual content, analyze key timestamps and clues, and then provide the final answer.

Video-MME – For other models

Select the best answer to the following multiple-choice question based on the video. Respond with only the letter (A, B, C, or D) of the correct option.
Question: {question} Possible answer choices:
{options}
Answer with the option's letter from the given choices directly.

MVBench – For other models

{question}
{options}
Only give the best option.

LongVideoBench – For other models

{question}
{options}
Answer with the option's letter from the given choices directly.

VideoMMMU – For other models

**Perception & Comprehension:**
{question}
{options}
Please ignore the Quiz question in last frame of the video.

- - - - - - - - - - - - - - - - - - - - - - - - - - - - - - - - - - - - - - - - - - - - -

**Adaptation-multiple-choice(open-ended):**
You should watch and learn the video content. Then apply what you learned to answer the following multi-choice(open-ended) question. The image for this question is at the end of the video.
{question}
{options}

---

MMVU – For Qwen series models and InternVL3.5-8B

**multiple-choice:**
{question}
{options}
Visual Information: processed video
Answer the given multiple-choice question step by step. Begin by explaining your reasoning process clearly. Conclude by stating the final answer using the following format: "Therefore, the final answer is: $LETTER" (without quotes), where $LETTER is one of the options. Think step by step before answering.

- - - - - - - - - - - - - - - - - - - - - - - - - - - - - - - - - - - - - - - - - - - - -

**open-ended:**
{question}
Visual Information: processed video
Answer the given question step by step. Begin by explaining your reasoning process clearly.
Conclude by stating the final answer using the following format: "Therefore, the final answer is: $ANSWER" (without quotes), where $ANSWER is the final answer of the question. Think step by step before answering.

---

MMVU – For other models

**multiple-choice:**
{question}
{options}
Visual Information: processed video
Do not generate any intermediate reasoning process. Answer directly with the option letter from the given choices.

- - - - - - - - - - - - - - - - - - - - - - - - - - - - - - - - - - - - - - - - - - - - -

**open-ended:**
{question}
Visual Information: processed video
Do not generate any intermediate reasoning process. Directly output the final answer.

---

SciVideoBench – For Qwen series thinking models, GPT-5.4, Gemini 3 Pro, Claude Opus 4.5 and InternVL3.5-8B

{question}
{options}
Answer the given question step by step. Begin by explaining your reasoning process clearly.
Conclude by stating the final answer using the following format: "Therefore, the final answer is: $ANSWER" (without quotes), where $ANSWER is the final answer of the question. Think step by step before answering.

---

SciVideoBench – For Qwen series instruct models and other models

{question}
{options}
Answer with the option's letter from the given choices directly.

**VideoKR-Eval– For Qwen series thinking models, GPT-5.4, Gemini 3 Pro, Claude Opus 4.5 and InternVL3.5-8B**

**multiple-choice:**
{question}
{options}
Answer the given multiple-choice question step by step. Begin by explaining your reasoning process clearly. Conclude by stating the final answer using the following format: "Therefore, the final answer is: $LETTER" (without quotes), where $LETTER is one of the options. Think step by step before answering.

- - - - - - - - - - - - - - - - - - - - - - - - - - - - - - - - - - - - - - - - - - - - - - - - - -

**open-ended:**
{question}
Answer the given question step by step. Begin by explaining your reasoning process clearly.
Conclude by stating the final answer using the following format: "Therefore, the final answer is: $ANSWER" (without quotes), where $ANSWER is the final answer of the question. Think step by step before answering.

---

**VideoKR-Eval– For Qwen series instruct models and other models**

**multiple-choice:**
{question}
{options}
Answer with the option's letter from the given choices directly.

- - - - - - - - - - - - - - - - - - - - - - - - - - - - - - - - - - - - - - - - - - - - - - - - - -

**open-ended:**
{question}
Please answer the question using a single word or phrase.

# D. Experiment Results and Analysis

## D.1. Performance with Different Frames

*Table 10.* Detailed accuracy on general and knowledge-intensive video reasoning benchmarks for post-trained models across different input frames.

| Model | Release | Frames | General Video Reasoning | | | | Knowledge-Intensive Video Reasoning | | | | |
| | | | Video-MME | MVBench | LongVideoBench | Average | VideoMMMU | MMVU | SciVideoBench | VideoKR-Eval | Average |
|---|---|---|---|---|---|---|---|---|---|---|---|
| **Post-trained Qwen2.5-VL-7B-Instruct** | | | | | | | | | | | |
| VideoKR (SFT+RL) | 2026-05 | 16 | 56.6 | 66.6 | 57.0 | 60.1 | **52.6** | 59.2 | 27.3 | 37.7 | 44.2 |
| VideoKR (SFT+RL) | 2026-05 | 32 | 60.1 | 68.2 | 58.2 | 62.2 | 51.2 | 58.9 | 27.4 | 39.8 | 44.3 |
| VideoKR (SFT+RL) | 2026-05 | 64 | 64.0 | 68.6 | 58.9 | 63.8 | **52.6** | **60.6** | 29.8 | 40.0 | 45.8 |
| VideoKR (SFT+RL) | 2026-05 | 128 | **66.4** | **68.9** | **61.3** | **65.5** | 52.2 | 60.5 | **32.5** | **41.2** | **46.6** |
| **Post-trained Qwen3-VL-8B-Instruct** | | | | | | | | | | | |
| VideoKR (SFT+RL) | 2026-05 | 16 | 57.2 | 65.5 | 54.7 | 59.1 | 60.2 | 63.8 | 29.3 | 40.5 | 48.5 |
| VideoKR (SFT+RL) | 2026-05 | 32 | 61.7 | 65.7 | 56.6 | 61.3 | 60.7 | 64.7 | 30.7 | 42.3 | 49.6 |
| VideoKR (SFT+RL) | 2026-05 | 64 | 63.9 | 66.2 | 57.9 | 62.7 | 62.1 | 63.8 | 31.5 | 44.6 | 50.5 |
| VideoKR (SFT+RL) | 2026-05 | 128 | **67.8** | **67.0** | **61.5** | **65.4** | **63.0** | **64.8** | **32.8** | **45.3** | **51.5** |

## D.2. Ablation Studies

*Table 11.* Ablation studies on post-training data. All experiments use Qwen2.5-VL-7B-Instruct as the base model, with 128 input frames.

| Ablation Setting | General Video Reasoning | | | | Knowledge-Intensive Video Reasoning | | | | |
|---|---|---|---|---|---|---|---|---|---|
| | Video-MME | MVBench | LongVideoBench | Average | VideoMMMU | MMVU | SciVideoBench | VideoKR-Eval | Average |
| Qwen2.5-VL-7B-Instruct | 65.1 | 66.3 | 60.9 | 64.1 | 51.1 | 55.7 | 28.1 | 32.7 | 41.9 |
| *Skill-Oriented Data Composition* | | | | | | | | | |
| VR (Basic Reasoning) | 61.0 | 60.8 | 52.3 | $58.0_{-6.1}$ | 51.3 | 51.7 | 27.3 | 35.3 | $41.4_{-0.5}$ |
| VR + KV (Perception) | 61.5 | 60.0 | 53.7 | $58.4_{-5.7}$ | 51.0 | 52.3 | 26.1 | 35.9 | $41.3_{-0.6}$ |
| VR + KV + KVR (Full) | 60.6 | 60.9 | 53.4 | $58.3_{-5.8}$ | 50.6 | 51.6 | 30.7 | 36.8 | $42.4_{+0.5}$ |
| *Supervision Format* | | | | | | | | | |
| Direct Output | 65.6 | 62.8 | 55.8 | $61.4_{-2.7}$ | 45.8 | 51.8 | 24.1 | 35.9 | $39.4_{-2.5}$ |
| Chain-of-Thought (CoT) | 60.6 | 60.9 | 53.4 | $58.3_{-5.8}$ | 50.6 | 51.6 | 30.7 | 36.8 | $42.4_{+0.5}$ |
| *Comparison with Other SFT Corpora (SFT-only)* | | | | | | | | | |
| Video-R1-CoT-165k | 56.9 | 63.2 | 51.9 | $57.3_{-6.8}$ | 45.3 | 50.4 | 21.4 | 27.5 | $36.2_{-5.7}$ |
| OneThinker-SFT-340k | 59.1 | 66.7 | 55.8 | $60.5_{-3.6}$ | 45.7 | 52.2 | 25.5 | 29.7 | $38.3_{-3.6}$ |
| VideoRFT-CoT-102K | 61.8 | 62.9 | 54.2 | $59.6_{-4.5}$ | 47.3 | 50.3 | 24.0 | 32.1 | $38.4_{-3.5}$ |
| VideoKR-SFT-201K (Ours) | 60.6 | 60.9 | 53.4 | $58.3_{-5.8}$ | 50.6 | 51.6 | 30.7 | 36.8 | $42.4_{+0.5}$ |
| *Comparison with Other RL Corpora (RL-only)* | | | | | | | | | |
| Video-R1-260k | 64.8 | 67.5 | 58.7 | $63.7_{-0.4}$ | 51.3 | 54.9 | 26.9 | 33.1 | $41.6_{-0.3}$ |
| OneThinker-600k | 63.7 | 64.5 | 52.7 | $60.3_{-3.8}$ | 53.7 | 54.5 | 27.6 | 33.2 | $42.3_{+0.4}$ |
| VideoRFT-RL-310K | 65.2 | 67.9 | 58.3 | $63.8_{-0.3}$ | 50.9 | 55.7 | 29.0 | 33.5 | $42.3_{+0.4}$ |
| VideoAuto-R1-83K | 65.0 | 66.9 | 56.6 | $62.8_{-1.3}$ | 52.4 | 55.9 | 29.0 | 33.3 | $42.7_{+0.8}$ |
| VideoKR-RL-114K (Ours) | 64.2 | 65.7 | 55.3 | $61.7_{-2.4}$ | 51.8 | 56.2 | 29.6 | 34.5 | $43.0_{+1.1}$ |

## D.3. Case Study

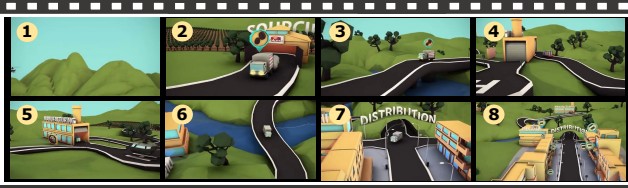

**Question:** For the company shown in the video, what is one of the methods to handle associated risks?
**Options:** (A) diversification (B) vertical integration (C) horizontal integration (D) outsourcing (E) speculation
**Explanation:** A coffee brand and its upstream chain are shown; to hedge green-bean price risk, the firm can vertically integrate—own/establish plantations to secure supply and stabilize costs.

Qwen3-VL-8B-Thinking: <think> The video showcases a supply chain process: procurement, manufacturing, distribution, and trading. Among the various options, there is no indication of diversification or speculation; nor is there any vertical or horizontal integration. Phased handover and **third-party transportation** imply outsourcing some processes to mitigate non-core risks. Therefore, option D (outsourcing) is the most suitable. </think><answer>D </answer>

**Error Reason:** Misread generic supply-chain stages as proof of outsourcing, ignoring cues (coffee beans, upstream cost volatility) that point to vertical integration as the risk-mitigation strategy. The correct inference is owning upstream production (e.g., plantations), not contracting it out.

OneThinker: <think> Video context: a supply-chain flowchart (sourcing → manufacturing → distribution → transactional management) indicates production and logistics stages. Option check: A) Diversification—portfolio risk spreading; not depicted. B) Vertical integration—end-to-end control; no evidence. C) Horizontal integration—mergers; absent. D) Outsourcing—**contracting external suppliers**; truck/boxed goods suggest third-party handling, letting the firm focus on core tasks. E) Speculation—risky bets; irrelevant here. Thus, the risk-handling method most consistent with the video is outsourcing. Final: D. </think><answer>D</answer>

**Error Reason:** Misread the scene as generic logistics and inferred outsourcing, ignoring that the bubbles indicate coffee beans and the focus is upstream cost volatility. For a coffee company, the correct risk mitigation depicted is vertical integration (own plantations), not outsourcing.

Qwen3-VL-8B-Instruct (SFT+RL on VideoKR): <think> The video depicts a truck departing from a coffee plantation adorned with coffee bean icons and the Folgers logo, heading towards a roasting/processing plant, and then to a retail store. The process traces the supply chain from SOURCING → MANUFACTURING → DISTRIBUTION. This highlights exposure to upstream green-coffee price and supply volatility. A fitting mitigation is vertical integration—**the brand establishes/owns plantations to secure supply**, stabilize input costs, and tighten quality control. Therefore, the answer is B, vertical integration. </think><answer>B</answer>

**Note:** From coffee-bean icons and the SOURCING → MANUFACTURING → DISTRIBUTION flow, it identifies upstream risk and correctly maps the mitigation to vertical integration (owning plantations), not outsourcing.

*Figure 9.* Comparison of model responses on a knowledge-intensive video reasoning sample.

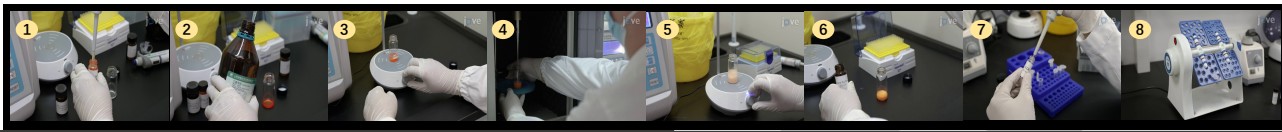

**Question:** What is the main purpose of the overnight magnetic stirring step starting at 02:31?

**Options:**
(A) Enhancing ultrasonication effects for smaller droplet size
(B) Increasing the solvent concentration for better dispersion
(C) Mixing to evenly distribute quantum dots in solution
(D) Preventing sedimentation of formed nanobeads
(E) Cooling the solution to induce polymer crystallization
(F) Breaking the emulsion into larger droplets
(G) Promoting chemical reaction between polymer and quantum dots
(H) Reducing temperature fluctuations during particle formation
(I) Removing excess water to concentrate the emulsion
(J) Complete evaporation of solvent leading to solidification

**Explanation:** This step is central to the emulsion–solvent evaporation route: ultrasonication (02:23) forms a chloroform emulsion, and overnight stirring renews the solvent–air interface to drive evaporation. As chloroform leaves, non-volatile PSMA and encapsulated QDs precipitate and harden into stable nanobeads; without it, solidification fails—thus the answer is J.

Qwen3-VL-8B-Thinking: <think> At 02:31 in the video, the procedure introduces an overnight magnetic stirring step during the preparation of QD-nanobead LFIA. In fact, prolonged gentle stirring helps homogenization, and more importantly, it keeps the particles suspended, preventing precipitation. The following options were evaluated: A (enhanced sonication), B (increased solvent concentration), E (cooling/crystallization), F (increased droplet size), G (promoted reaction), H (temperature control), and I/J (concentration/evaporation). None of these align with the typical purpose of stirring. C (uniform distribution of QD), while feasible, is less important than **maintaining suspension** —the core objective. Therefore, the primary purpose of this step is D: preventing the precipitation of the formed nanobeads. </think><answer>D</answer>

**Error Reason:** Misread the step as anti-sedimentation (D). The sequence shows a volatile-solvent emulsion and later dry beads, indicating overnight stirring primarily drives complete solvent evaporation and solidification (J).

OneThinker: <think> The video demonstrates an overnight magnetic stirring step. In laboratory practice, prolonged gentle stirring is used to **maintain the homogeneity of the dispersion**, prevent precipitation/aggregation, and improve mass transfer, thereby ensuring a uniform distribution of the suspension (in this case, quantum dots) in the next step. It is not used to change solvent concentration (B), cool (E), control temperature (H), concentrate/evaporate (I, J), or increase droplet size (F). Enhanced sonication (A) is also not applicable—stirring is a low-shear process, typically performed **after** sonication, and its role is to maintain droplet integrity, not to further break down the droplets. Preventing precipitation (D) and any accidental interactions (G) may occur, but they are secondary. Therefore, the purpose is to mix to ensure a uniform distribution of the quantum dots in the solution—option C. </think><answer>C</answer>

**Error Reason:** Misread overnight stirring as simple homogenization (C), ignoring cues of a volatile-solvent emulsion and later dry beads. In this workflow, stirring primarily drives complete solvent evaporation and solidification (J), not just preventing settling.

Qwen3-VL-8B-Instruct (SFT+RL on VideoKR): <think> The video at 02:31 demonstrates the overnight magnetic stirring step in the nanobead preparation process. Overnight magnetic stirring is a routine homogenization step in the quantum dot-nanobead preparation process, and is therefore likely for maintaining uniform dispersion (C) and inhibiting precipitation (D). Comparing other options, many (B, E, H, I) do not conform to the typical function of stirring; D seems best suited for the overnight step. **Wait**— rewatching the segment around 02:23, we can see ultrasonic treatment **forming an emulsion** in a volatile solvent (chloroform), followed by footage of dried, solid nanobeads. During the emulsion-solvent evaporation process, the prolonged gentle stirring primarily serves to renew the solvent-air interface between droplets, **thereby promoting complete solvent evaporation**, allowing the polymer + quantum dots to precipitate and harden. Therefore, preventing precipitation is secondary; the primary goal is complete evaporation, ultimately leading to solidification—J. </think><answer>J</answer>

**Note:** Initial reasoning favored homogenization/anti-settling (C/D), but the "aha moment" came when linking the ultrasonication-created chloroform emulsion (02:23) to the later dry beads—revealing that overnight stirring chiefly drives complete solvent evaporation and solidification (J).

*Figure 10.* Comparison of model responses on a knowledge-intensive video reasoning sample.

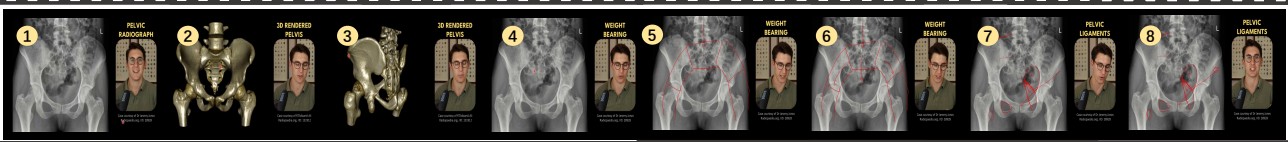

**Question:** Can you identify the abnormality on this plain film of the pelvis?
<image 1>
**Options:**
(A) Bone cyst
(B) Acute hip fracture
(C) Osteoarthritis
(D) Surgical hardware
(E) Resection of the pubic symphysis
(F) Bone tumor
(G) Hip dislocation
(H) Osteoporosis
(I) Paget's disease
(J) Bone infection

**Explanation:** The key finding is a smooth cortical bone defect at the pubic symphysis, suggesting a previous pubic symphysis resection rather than acute separation or trauma. The sacroiliac and hip joints are well-matched, with no fracture lines or metal implants, and no signs of osteoarthritis, Paget's disease, osteoporosis, infection, or tumor.

**Qwen3-VL-8B-Thinking:** <think> The video shows a diffuse radiolucent area in the pelvis, accompanied by widespread cortical thinning and loss of normal trabecular detail. The hip joint is symmetrical, without displacement, and no focal osteolytic or sclerotic lesions, fracture lines, periosteal reaction, or metallic density shadows are observed. The joint space is intact, with no osteophyte formation, suggesting the absence of osteoarthritis, and lacks the typical features of Paget's disease (bone expansion, cortical thickening) or infection (invasive destruction, soft tissue swelling). The lesion is diffuse rather than focal, therefore osteoporosis is the most plausible explanation. </think><answer>H</answer>

**Error Reason:** The model incorrectly overinterpreted diffuse radiolucent areas as osteoporosis, neglecting a crucial finding—smooth cortical spaces at the pubic symphysis, consistent with surgical resection. The apparent "demineralization" may be related to bone protrusion/exposure, and there is no corroborating evidence of osteopenia prevalent in other sites.

**OneThinker:** <think> Based on the pelvis plain film and the listed options, the abnormality most plausibly involves the hip joint. Typical signs of a bone cyst, tumor, osteoporosis, Paget's disease, osteoarthritis, pubic symphysis resection, hip dislocation, or bone infection are not clearly demonstrated, and an acute hip fracture is not convincingly visible. With the absence of definitive features for these alternatives, the most specific and common explanation in this context is a hip-related abnormality consistent with surgical hardware. Therefore, the best answer is D. Surgical hardware. </think><answer>D</answer>

**Error Reason:** Although no metallic density was observed on the imaging, the model incorrectly inferred the presence of a "surgical implant" and overlooked the smooth pubic symphysis and cortical absence—characteristic features of previous resections. Furthermore, the diagnosis incorrectly described the abnormality as related to the hip joint, rather than located at the pubic symphysis.

**Qwen3-VL-8B-Instruct (SFT+RL on VideoKR):** <think> The video shows **significant destruction at the pubic symphysis**, with a marked gap or separation, which is uncommon in a normal pelvis. Therefore, there is a significant abnormality in the pubic symphysis region. The **interpubic cortical space is smooth**, and the surrounding bone structure appears intact, **but this is consistent with previous surgical resection rather than acute trauma.** The hip joint is well-aligned, without displacement or dislocation; no cortical fracture or trabecular bone destruction is observed, nor is there an acute hip fracture; no focal osteolytic/sclerotic lesions, bone cysts, tumors, Paget's disease, or infection are seen. Diffuse demineralization suggests osteoporosis, but is insufficient for a definitive diagnosis; typical features of osteoarthritis such as joint space narrowing or osteophyte formation are also absent. No metallic feel or surgical implants are observed. Therefore, the pathological findings are most consistent with a diagnosis following pubic symphysis resection. </think><answer>E</answer>

**Note:** VideoKR focuses on the pubic symphysis and utilizes radiological prior information (depicting the pelvic ring and checking joint consistency) to avoid the biases of other models that are hip-centric and diffuse osteoporosis.

*Figure 11.* Comparison of model responses on a knowledge-intensive video reasoning sample.

