# OpenReview forum: "VideoKR: Towards Knowledge- and Reasoning-Intensive Video Understanding"
_ICML.cc/2026/Conference — ICML 2026 spotlight_

### Official Review · Reviewer_NNQL · 2026-02-22

**Soundness:** 4
**Presentation:** 4
**Significance:** 4
**Originality:** 4
**Overall Recommendation:** 6
**Confidence:** 5

**Summary:**

This paper proposes a novel Knowledge- and Reasoning-Intensive video post-training dataset (including cold start SFT and RL), which is constructed through elaborately designed procedures such as video source collection and video QA generation. Its effectiveness is validated on a wide range of widely used video benchmarks. In addition, to address the limitations of existing video knowledge benchmarks, this paper further presents VideoKR-Eval, which offers a more reliable evaluation of video knowledge and reasoning capabilities.

**Compliance With Llm Reviewing Policy:**

Affirmed.

**Key Questions For Authors:**

See Weakness.

**Limitations:**

See Weakness.

**Strengths And Weaknesses:**

### Strengths

1. In response to the scarcity of existing academic training data related to video knowledge, the training set proposed in this paper is collected and constructed entirely from scratch. The designed pipeline for building the training data is highly meticulous, and sufficient experimental results are provided to verify its reliability. It is believed that open-sourcing this dataset will greatly benefit the research community.

2. The proposed VideoKR-Eval effectively identifies the shortcomings of existing evaluation benchmarks. Meanwhile, rigorous manual re-inspection and annotation enable it to serve as a new and reliable benchmark.

### Weakness

1. For the training data, further analytical experiments would be beneficial: The thinking trace of VideoKR-SFT relies on distillation from state-of-the-art models. It would be worthwhile to compare the quality between the cold-start data and the cold-start training data rolled out by VideoKR (zero RL), which could further reveal the quality of the VideoKR training query set.

2. As shown in Table 5, VideoKR exhibits higher difficulty than many previous training datasets. For the RL data, a subset further filtered by difficulty may be more suitable for RL training. I suggest supplementing its performance into Table 4.

---

> ### Author Rebuttal · Authors · 2026-03-31
>
> We sincerely thank you for the time and constructive feedback. We would like to address your concerns and questions as follows:
>
> ---
>
> > **W1:** For the training data, further analytical experiments would be beneficial: The thinking trace of VideoKR-SFT relies on distillation from state-of-the-art models. It would be worthwhile to compare the quality between the cold-start data and the cold-start training data rolled out by VideoKR (zero RL), which could further reveal the quality of the VideoKR training query set.
> >
>
> Thank you for this insightful suggestion. We conducted the proposed controlled comparison as follows: we re-rolled the 80K SFT queries using the VideoKR zero-RL model, retaining only samples with correct final answers, which yielded a matched 41K subset. We then trained on two aligned versions of this subset: (1) the original VideoKR-SFT annotations (distilled from frontier models and refined through our quality-control pipeline), and (2) the zero-RL-rolled annotations.
> Results are as follows:
>
> | Variant | Knowledge-Intensive Avg |
> | --- | --- |
> | Base model (Qwen2.5-VL-7B-Instruct) | 39.8 |
> | Zero-RL-rolled annotations | 40.6 (+0.8) |
> | Original VideoKR-SFT annotations | 42.4 (+2.6) |
>
> Two conclusions emerge:
>
> - The original SFT traces outperform the zero-RL-rolled ones (42.4 vs. 40.6), which is expected since they are distilled from stronger frontier models and undergo multi-stage validation (self-consistency check, video dependency filtering, and CoT rationale validation; §3.3+3.4)
> - Perhaps more importantly, even the zero-RL-rolled traces yield clear gains over the base model (+0.8), demonstrating that the VideoKR QA set itself carries substantial value: its knowledge-grounded, skill-oriented questions are informative enough to elicit useful reasoning even from a weaker generator
>
> &nbsp;
>
> > **W2:** As shown in Table 5, VideoKR exhibits higher difficulty than many previous training datasets. For the RL data, a subset further filtered by difficulty may be more suitable for RL training. I suggest supplementing its performance into Table 4.
> >
>
> We appreciate this suggestion and agree that VideoKR's higher difficulty (Table 5) naturally motivates a difficulty-stratified RL setup. We conducted the following analysis: using Qwen3-VL-8B-Instruct, we performed 5 independent rollouts per example on VideoKR-RL-170K and categorized difficulty by the number of correct rollouts:
>
> | Difficulty | Criterion | Proportion |
> | --- | --- | --- |
> | Easy | 4, 5 / 5 correct | 24% |
> | Medium | 2, 3 / 5 correct | 37% |
> | Difficult | 0, 1 / 5 correct | 39% |
>
> Building on this, we are currently running two sets of experiments:
>
> - Following your suggestion, we conduct difficulty-filtered RL ablations. Starting from the SFT checkpoint, we GRPO-train on three 50K-example variants randomly sampled from VideoKR-RL-170K: easy-removed, difficult-removed, and medium-only.
> - As an additional exploration, we test a curriculum-style RL schedule (easy → medium → difficult) to examine whether structured difficulty progression yields further gains over uniform mixing. We also note that several works in other domains are exploring related strategies, which we will discuss in the future work section.
>
> RL training is computationally intensive, these experiments are still in progress and currently queued on our GPU cluster. We will include the full results in the final revision.

---

> > ### Author Rebuttal · Reviewer_NNQL · 2026-04-04
> >
> > Thank the authors for their detailed rebuttal. I maintain my support for acceptance of this paper.

---

### Official Review · Reviewer_MQY4 · 2026-03-10

**Soundness:** 3
**Presentation:** 2
**Significance:** 3
**Originality:** 3
**Overall Recommendation:** 6
**Confidence:** 4

**Summary:**

This paper introduces VideoKR, a large-scale video dataset designed specifically for knowledge-intensive reasoning tasks. The core contribution is shifting the paradigm from complex reward engineering in video reasoning models toward prioritizing high-quality data curation. This paper also conducts comprehensive ablations to isolate the contributions of VideoKR.

**Compliance With Llm Reviewing Policy:**

Affirmed.

**Final Justification:**

The authors have addressed my concerns.

**Key Questions For Authors:**

see the weakness

**Limitations:**

see the weakness

**Strengths And Weaknesses:**

## Strength

- The paper addresses a critical bottleneck in video understanding by introducing VideoKR, a large-scale corpus of 126K CC-licensed videos and 430K QA pairs. Shifting the focus from complex reward engineering to data quality is a practical and highly impactful perspective for the community.

- The data construction methodology is rigorous and technically sound. The use of a hierarchical domain knowledge bank covering 82 subjects ensures broad and structured coverage. Furthermore, strict filtering mechanisms, such as video dependency checks and CoT rationale validation, ensure the dataset genuinely requires complex visual reasoning.

- The evaluation framework is well-designed. By using a standard SFT to GRPO pipeline, the authors successfully isolate the impact of their dataset from confounding algorithmic complexities.  Additionally, the paper conducts comprehensive ablation studies, fully validating the effectiveness of the proposed VideoKR-SFT-260K and VideoKR-RL-170K datasets.

## Weakness

- The dataset construction explicitly filters out videos exceeding 30 minutes. Since many real-world, knowledge-intensive scenarios (such as full academic lectures or complex surgical procedures) inherently require ultra-long-context understanding, this exclusion limits the method's ability to address the full spectrum of expert-level video reasoning.

- The authors employ rigorous visual decontamination methods, including YouTube ID matching and perceptual frame hashing, to prevent evaluation leakage. However, there is a lack of discussion regarding textual decontamination; since the dataset focuses on domain knowledge and uses LLMs for QA generation, there is a risk that the textual queries or factual answers may overlap with the evaluation sets or the base models' pre-training data.

- This paper misses some references about video datasets, such as videovista

- Some RFT technicals for video reasoning should be dicussed and the challenging future directions should be presented in this paper.

---

> ### Author Rebuttal · Authors · 2026-03-31
>
> We sincerely thank you for the thoughtful evaluation of our paper and the constructive feedback. We would like to address your concerns as follows:
>
> > **W1:** The dataset construction explicitly filters out videos exceeding 30 minutes, …. limits the method's ability to address the full spectrum of expert-level video reasoning
>
> Thanks for your thoughtful comment. The 30-minute cap was primarily a practical decision driven by current base model capabilities. As base model’s effective context windows continue to scale, extending VideoKR to longer videos is a natural and straightforward next step.  Encouragingly, our pipeline is already well suited to such future extension. The core stages, including knowledge-driven video collection (§3.2), skill-oriented QA generation (§3.3), and expert-validated quality control, do not depend on video duration. In practice, supporting videos beyond 30 minutes would mainly require relaxing the filtering threshold, without changing the pipeline design.
>
> At the same time, as shown in Table 1, VideoKR already substantially extends video duration relative to existing post-training corpora. Moreover, through our knowledge-driven video collection stage (§3.2), the video corpus already covers a broad range of real-world expert scenarios.
>
> We also note that knowledge-intensive reasoning and ultra-long-context understanding tend to pose orthogonal challenges: The reasoning depth that VideoKR targets (i.e., multi-hop inference grounded in domain knowledge and visual evidence) often unfolds within localized temporal windows even in longer videos. Our results on LongVideoBench (Table 3) suggest that the current duration distribution supports strong generalization to longer video understanding.
>
> We will clarify these point in the revision.
>
> &nbsp;
>
> > **W2:** The authors employ rigorous visual decontamination methods… However, there is a lack of discussion regarding textual decontamination; …, there is a risk that the textual queries or factual answers may overlap with the evaluation sets or the base models' pre-training data
>
> Thanks for raising this important point. We have added two new analyses to address the concern about textual contamination:
>
> - We extracted the textual content from 61 video-related benchmarks supported by LMMs-Eval and conducted an exhaustive example-level comparison using 9-gram matching against 100K randomly sampled questions from VideoKR. To scale this search, we ran the matching pipeline with 256 parallel processes. Under this criterion, only 73 of the 100K sampled VideoKR examples exhibited any 9-gram overlap. We then manually examined these cases and confirmed that every one of these was a false positive (e.g., shared boilerplate phrasing rather than duplicated questions). This outcome is consistent with expectation: our visual decontamination step makes textual leakage exceedingly unlikely by design.
> - Since Qwen’s pre-training corpus is not publicly available, direct decontamination against the pre-training data is infeasible. To instead test whether memorized textual knowledge alone could explain performance on VideoKR, we evaluate Qwen3-VL-8B-Instruct on the textual questions only, without access to video frames. We randomly sample 5,000 multiple-choice training examples from VideoKR. The model achieves 29.7% accuracy, which is close to random chance (25%) and substantially below the 46.8% accuracy with video+text input (reported in Table 5).
>
> &nbsp;
>
> > **W3:**  This paper misses some references about video datasets, such as videovista
>
> Thank you for pointing this out. We have carefully revisited the recent literature. In the revised manuscript, we added **VideoVista** and other concurrent benchmark works (i.e., **ExpVid, RIVER, MSVBench, VideoReasonBench**) to the related work section. We would appreciate any further recommendations.
>
> &nbsp;
>
> > **W4:**  Some RFT technicals for video reasoning should be discussed and the challenging future directions should be presented in this paper
>
> Thank you for the valuable suggestions. We provide a detailed discussion of the technical aspects of RFT for video reasoning in Appendix A. In the revised manuscript, we also revisited the recent literature and added discussion of several recent works (i.e., **VerIPO, Video-KTR, ReWatch-R1, and LLMs Meet Long Video**). We will continue to monitor relevant developments.
>
> Regarding challenging future directions, we also added a new Future Work section. In brief, we discuss:
>
> - The skill-oriented data composition ablation (Table 4) reveals a clear stepping-stone effect, suggesting that RL training may benefit from a curriculum that progressively increases knowledge and reasoning demands
> - §4 reveals a systemic issue in existing benchmarks, underscoring the urgent need for next-generation benchmarks that rigorously enforce visual dependency
> - Exploring process-level reward design for knowledge-intensive video reasoning
> - Extending VideoKR to the long-video setting

---

> > ### Author Rebuttal · Reviewer_MQY4 · 2026-04-02
> >
> > Thank you for your reply. My concern has been basically solved, and I hope that the author adds the relevant content to the next camera-ready version as promised. I will promote the rating score.

---

### Official Review · Reviewer_Ccoe · 2026-03-13

**Soundness:** 3
**Presentation:** 3
**Significance:** 3
**Originality:** 3
**Overall Recommendation:** 4
**Confidence:** 4

**Summary:**

This paper introduces VideoKR, a large-scale dataset designed to improve knowledge-intensive and reasoning-intensive video understanding. The authors argue that current video-language models mainly focus on surface-level perception tasks (e.g., object recognition or simple temporal reasoning) and lack the ability to perform deeper reasoning that combines video evidence with external knowledge. To address this limitation, the paper constructs VideoKR, a dataset containing 430K reasoning examples from 126K videos. The dataset is generated through an expert-in-the-loop pipeline that produces reasoning-oriented question–answer pairs with chain-of-thought rationales. The pipeline emphasizes skill-oriented data generation to target progressively more complex reasoning capabilities. The authors also introduce VideoKR-Eval, a curated benchmark designed to evaluate knowledge-based video reasoning. Models are post-trained using a SFT → GRPO pipeline on the VideoKR dataset. Experimental results show that models trained on VideoKR outperform previous post-training approaches on several video reasoning benchmarks, demonstrating the importance of carefully designed reasoning-oriented data.

**Compliance With Llm Reviewing Policy:**

Affirmed.

**Final Justification:**

Increased my score to 4 since rebuttal fully addressed my concerns

**Key Questions For Authors:**

please see my comments in waeknesses

**Limitations:**

This paper did not include a limitation section in the submission.

**Strengths And Weaknesses:**

Strengths:

1. VideoKR is a sizable dataset (430K examples) specifically designed to target knowledge-intensive video reasoning, which is currently an active field like video-mmmu, video-mmlu, scivideobench and expvid.

2. The proposed expert-in-the-loop pipeline attempts to generate examples with increasing reasoning complexity, which is a thoughtful design for constructing higher-quality reasoning data.

3. Experiments show that post-training models on VideoKR improves performance on both general video reasoning tasks and knowledge-intensive benchmarks.

4. The work highlights the importance of dataset construction and reasoning supervision, which may provide useful guidance for future video-language model training.


Weaknesses:

1. The main novelty lies in dataset construction rather than new modeling techniques. The training framework (SFT + GRPO) largely follows existing approaches.

2. Although the dataset includes chain-of-thought rationales, the paper provides limited analysis of whether the models genuinely perform deeper reasoning or simply learn dataset patterns.

3. Some tasks might be solvable using prior knowledge or textual cues without truly understanding the video content, which raises questions about how strongly the tasks require visual grounding.

4. The pipeline involves expert participation and reasoning annotation, but the paper does not thoroughly discuss the cost or scalability of this process.

5. The expert-in-the-loop generation pipeline may introduce biases or artifacts in the reasoning patterns, which could limit generalization.

---

> ### Author Rebuttal · Authors · 2026-03-31
>
> We sincerely thank you for the time and constructive feedback. We would like to address your concerns as follows:
>
> > **W1:** The main novelty lies in dataset construction rather than new modeling techniques. The training framework (SFT + GRPO) largely follows existing approaches
>
> Thanks for your thoughtful comments. We wish to clarify that prioritizing data-centric analysis is a deliberate methodological choice. As noted in §5.1, we aim to test whether data design is a key limiting factor for video reasoning, a question that has been comparatively underexplored. To isolate this factor, we intentionally adopt a standard SFT→GRPO pipeline as a controlled training scaffold, allowing the observed improvements to be attributed more directly to the training data. Our results show that, even with this standard setup, our trained model surpasses previous post-training methods across all eight benchmarks (Tables 3&4). This highlights a gap that better data can fill alongside advances in algorithmic novelty.
>
> Moreover, our pipeline itself contributes reusable methodological innovations: the hierarchical knowledge bank, skill-oriented decomposition, multi-model generation with human-validated per-step eligibility, and video dependency filtering stricter than existing benchmark standards. We view these contributions as complementary to future algorithmic advances in video reasoning and as potentially amplifying their benefits.
>
> &nbsp;
>
> > **W2: …** the paper provides limited analysis of whether the models genuinely perform deeper reasoning or simply learn dataset patterns
>
> Thanks for your thoughtful comments. The paper provided several analyses:
>
> - Our trained models show improvements across all six external benchmarks (Table 3), whose videos and question distributions differ substantially
> - Our Inference-time frame scaling analysis (§6.3) shows consistent performance gains as frames increase from 16 to 128. If the model memorized dataset patterns, performance would be largely independent of the number of input frames at inference.
> - We present qualitative examples (§6.2, Appendix F.3) where the VideoKR-trained model initially considers incorrect hypotheses, re-examines video evidence, and self-corrects. Such reflective, evidence-grounded behavior is less consistent with simple pattern matching
> - Our training-data difficulty analysis (§6.5) shows that VideoKR provides genuinely more challenging learning signals than prior corpora.
>
> &nbsp;
>
> > **W3:** Some tasks might be solvable using prior knowledge or textual cues without truly understanding the video content, which raises questions about how strongly the tasks require visual grounding.
>
> This is precisely the issue we identify and address. In §4.1, we reveal the issue of “non-visually solvable” in existing benchmark. It motivates several design choices:
>
> - Our scenario-based video search strategy (§3.2) retrieves videos where knowledge is implicitly applied rather than explicitly explained. It ensures that reasoning requires observing and interpreting the visual content, not just recognizing textual keywords
> - We apply a stricter filtering procedure (§3.3) than prior work to remove examples that can be solved without actual video understanding
> - We re-annotate existing benchmarks to construct VideoKR-Eval (§4.2), which exhibits a substantially lower non-visual solvability rate and therefore provides a more reliable test of visual grounding
>
> &nbsp;
>
> > **W4**: The pipeline involves expert participation and reasoning annotation, but the paper does not thoroughly discuss the cost or scalability of this process.
>
> We reported cost details in the Impact Statements section. Regarding scalability, several pipeline components are explicitly designed to support future expansion:
>
> - Example generation is automated using frontier models, with human experts primarily acting as quality gatekeepers
> - Human-validated model selection is a one-time cost (§3.4)**.** The same eligibility assignments are reused throughout large-scale synthesis
> - Self-consistency checks, video dependency filtering, and CoT rationale validation are all model-based, which substantially reduces human involvement to final spot-checking (§3.4)
>
> &nbsp;
>
> > **W5:** The expert-in-the-loop generation pipeline may introduce biases or artifacts in the reasoning patterns…
>
> Thanks for your thoughtful comments. We implemented several mechanisms to mitigate such biases:
>
> - Unlike prior work that relies on a single model (Table 1), we maintain a pool of seven frontier models for example generation, which reduces any single model's systematic artifacts
> - The self-consistency check (§3.3) re-derives answers independently, and the CoT rationale validation uses a different model than the generator, to mitigate generator-specific biases
> - Our annotators have diverse academic backgrounds (Table 7)
> - The seed examples are independently reviewed by a second annotator (§3.3)
> - We conduct careful data contamination mitigation (§3.4)

---

> > ### Author Rebuttal · Reviewer_Ccoe · 2026-03-31
> >
> > Thanks for clarification. My concerns have been adequately addressed

---

### Official Review · Reviewer_87CC · 2026-03-13

**Soundness:** 4
**Presentation:** 3
**Significance:** 3
**Originality:** 3
**Overall Recommendation:** 4
**Confidence:** 4

**Summary:**

This paper presents a large-scale video reasoning dataset to enable post-training of multimodal LLMs on video reasoning. Videos are first sampled from YouTube focussing on extracting videos related to pedagogical scenarios such as Newton's laws. Then, multiple-choice questions and open-ended questions are created using multimodal large-language models seeded from expert annotations. The generated data is quality checked by automated LLM checks such as CoT validation, visual dependency etc., and a small sample is validated with human experts. Furthermore, a reasoning focused evaluation set with a cleaner human verified ground-truth is proposed. Experimental results are presented to demonstrate the impact of post-training on the proposed dataset.

**Compliance With Llm Reviewing Policy:**

Affirmed.

**Final Justification:**

Rebuttal has addressed my concerns and I retain my recommendation of Weak Accept.

**Key Questions For Authors:**

Please check above section.

**Limitations:**

Limitations section is not provided.

**Strengths And Weaknesses:**

## Strengths:
- The paper is well-written and easy to follow.
- The paper presents a large-scale video reasoning benchmark for post-training, which is a core component of multimodal LLM training. Improving video reasoning during post-training can help solve a variety of downstream video understanding applications.
- The paper takes a methodological approach to video selection, starting from subjects --> courses --> lectures --> knowledge points, which are then turned into search queries resulting in a diverse coverage.
- The paper avoids many common pitfalls in (semi-)automated data generation using MLLMs:
  - A variety of LLMs are used to avoid biases
  - Reasoning process is seeded from human chain-of-thought from the same domain.
  - QAs answerable from text & few-frames are eliminated. This makes sure the QAs test video understanding.
- A new benchmark is presented, and instead of constructing yet another benchmark from scratch, existing benchmarks are aggregated and de-noised manually. This is a welcome step allowing for greater diversity and assurance of quality.
- Experimental results in Table1 clearly show the generalizability; model post-trained on proposed data is doing well across general video understanding and reasoning.
- Table 4 ablation is also quite welcome in comparing post-training on the proposed data against post-training on other competitor datasets. Results show the superiority of the method.
- The data is promised to be fully open-sourced.

## Weaknesses:
- The paper claims that it is the first large-scale training corpus for post-training on video reasoning. This does injustice to prior works [1] and well-known concurrent works. [2]
- In Table 1, it seems like the proposed dataset is a way to distill information from closed-source and larger models into a smaller model. At the end of the day, the data is generated from the closed-source and larger models.
- Another interesting question that's not addressed is whether we actually need a video reasoning dataset or whether learning video understanding can be decoupled from learning reasoning.
  - Table 1 offers some insights that blindly scaling without integrating the proposed dataset is also quite beneficial for video reasoning.
  - There is a big gap between closed source model performance and smaller models post-trained with the proposed dataset.
  - It would be interesting to see any post-training results at larger scale to verify this.

[1] VideoEspresso: A Large-Scale Chain-of-Thought Dataset for Fine-Grained Video Reasoning via Core Frame Selection \
[2] A Very Big Video Reasoning Suite

---

> ### Author Rebuttal · Authors · 2026-03-31
>
> We sincerely thank you for the time and constructive feedback. We would like to address your concerns and questions as follows:
>
> > **W1:** The paper claims that it is the first large-scale training corpus for post-training on video reasoning. This does injustice to prior works and well-known concurrent works.
>
> We appreciate the opportunity to clarify. In the abstract and introduction introduction, our claim is specifically scoped to "the first large-scale training corpus specifically designed to strengthen knowledge- and reasoning-intensive video understanding".
>
> We respect the contributions of prior and concurrent works and provided a detailed comparison with these efforts in Table 1 and the related work section, which already included the VideoEspresso[1] work you mentioned. [2] was released after our submission, we will discuss it in the revised manuscript. We will continue to monitor relevant developments and would appreciate any further recommendations.
>
> &nbsp;
>
> > **W2:** In Table 1, it seems like the proposed dataset is a way to distill information from closed-source and larger models into a smaller model. At the end of the day, the data is generated from the closed-source and larger models.
> >
>
> Thanks for your insightful comments. As shown in Table 1, model-assisted data synthesis is widely adopted across prior post-training corpora for video understanding. Our approach is different from distillation, where a teacher's outputs on a fixed set of inputs are directly used as supervision. In our work, we focus on designing a multi-stage, quality-controlled pipeline for generating high-quality video reasoning QA data, grounding supervision in newly collected CC-licensed videos, domain knowledge bank, expert-curated seed examples, and multiple layers of validation and filtering including human-validated model selection across a pool of seven frontier models. The frontier models serve as tools within this pipeline, not as teachers whose behavior is to be imitated. Notably, in the zero-RL setting, where training starts directly from Qwen3-VL-8B-Instruct without any SFT or teacher reasoning rationale, we still observe substantial gains (Table 3).
>
> &nbsp;
>
> > **W3.1:** Another interesting question that's not addressed is whether we actually need a video reasoning dataset or whether learning video understanding can be decoupled from learning reasoning.
>
> Thank you for raising this important question. Our experiments provide two lines of evidence that integrating domain knowledge and reasoning into video training data is critical
>
> - Joint training across skill levels is essential. Our skill-oriented data generation pipeline uniquely enables us to isolate the contribution of each capability layer. The skill-composition ablation (§6.4) demonstrates a clear stepping-stone effect: training on Basic Video Reasoning alone yields 42.2% on knowledge-intensive benchmarks; adding Knowledge-Enhanced Perception raises this to 42.8%; and further incorporating Knowledge-Intensive Reasoning reaches 43.9%.
> - Existing benchmarks may overstate the benefits of decoupled approaches. As shown in Table 2, frontier LLMs can answer over 40% of questions on MMVU and VideoMMMU without any visual input, inflating scores for methods that primarily improve text-based reasoning. On our expert-reannotated VideoKR-Eval, which enforces genuine visual dependency, our models show substantially larger gains (e.g., +6.6 points for Qwen3-VL-8B), suggesting that tightly coupling visual understanding with reasoning is especially important when evaluation truly requires both.
>
> &nbsp;
>
> > **W3.1:** Table 1 offers some insights that blindly scaling without integrating the proposed dataset is also quite beneficial for video reasoning.
>
> In our controlled ablation study (§6.5), under the same 80K-example, one-epoch SFT setting, VideoKR achieves 43.9% on knowledge-intensive benchmarks, compared with 39.3% for Video-R1-CoT and 38.8% for VideoRFT-CoT. The latter actually falling below the 39.8% base model. This suggests that prior corpora are insufficient for knowledge-intensive video reasoning, and the composition and design of training data matters more than volume alone.
>
> &nbsp;
>
> > **W3.2:** There is a big gap between closed source model performance and smaller models post-trained with the proposed dataset. **W3.3:** It would be interesting to see any post-training results at larger scale to verify this.
>
> We agree that a substantial gap remains. We view this as a broader challenge for the open-source community. Our goal is not to claim that VideoKR alone can fully close this gap, but to provide a high-quality training resource that serves as a complementary ingredient for narrowing it.
>
> Following your suggestions, we added a larger-scale experiment using Qwen3-VL-32B-Instruct with the same setup as Section 6.5 (SFT on 80K examples for one epoch). We observe consistent improvements over the base 32B model, from 58.9 to 61.4 on knowledge-intensive average.

---

> > ### Author Rebuttal · Reviewer_87CC · 2026-04-04
> >
> > I would like to thank the authors for the detailed rebuttal

---

### Decision · Program_Chairs · 2026-04-30

**Decision:**

Accept (spotlight)

**Comment:**

This paper receives: weak accept, weak accept, strong accept, strong accept. The reviewers all appreciate the contribution of this paper on collecting reasoning intensive video data for post-training and evaluation. The experiments demonstrate the effectiveness of the newly-collected data and the value of the built benchmark. Thus, the AC agrees with reviewers and makes an accept recommendation. The authors are encouraged open-source the training data and benchmark to facilitate the future research.